# Value Factorization for Asynchronous Multi-Agent Reinforcement Learning

## Abstract

Value factorization has become widely used to design high-quality and scalable multi-agent reinforcement learning algorithms. However, existing methods assume agents execute synchronously, which does not align with the asynchronous nature of real-world multi-agent systems. In these systems, agents often make decisions at different times, executing asynchronous (*macro-*)actions characterized by varying and unknown duration. Our work introduces value factorization to the asynchronous framework. To this end, we formalize the consistency requirement between joint and individual macro-action selection, proving it generalizes the synchronous case. We then propose approaches that use asynchronous centralized information to enable factorization architectures to support macro-actions. We evaluate the resultant asynchronous value factorization algorithms across increasingly complex domains that are standard benchmarks in the macro-action literature. Crucially, the proposed methods scale well in these challenging coordination tasks where their synchronous counterparts fail.

## 1 Introduction

Multi-agent reinforcement learning (MARL) has achieved impressive results in several cooperative domains (Samvelyan et al., 2019; Mordatch & Abbeel, 2017). Due to partial observability and communication constraints, MARL algorithms often aim to learn policies that condition on local information, while leveraging centralized data at training time to foster collaborative behaviors (i.e., *Centralised Training with Decentralised Execution* (CTDE) (Tuyls & Weiss, 2012)). Value-based algorithms have been particularly successful at CTDE by factoring a joint action value into per-agent utilities conditioned on local information (Rashid et al., 2018; Wang et al., 2021). To achieve a sound factorization, these methods ensure consistency between the local greedy actions and the joint greedy action selection; a principle known as the *Individual Global Max* (IGM) (Son et al., 2019). Agents can thus execute in a decentralized manner by selecting actions according to the local utilities while learning in a centralized fashion using a factored, joint action-value.

Despite the merits of factorization, current algorithms assume agents execute in a *synchronous* way (Yu et al., 2022; Rashid et al., 2018; Wang et al., 2021)—selecting a *primitive action* that starts and ends simultaneously for each agent at each execution step. This assumption does not hold in many real-world scenarios where agents use *asynchronous decision-making*, selecting and completing actions with varying duration at different times. These temporally extended behaviors are known as *macro-actions* and generalize the primitive action case, allowing agents to be asynchronous (Amato et al., 2019). Macro-actions have several advantages over primitive ones as they (i) foster explainability by representing complex real-world behavior that may require multiple steps (e.g., navigating to a waypoint, waiting for a human); (ii) benefit value-backup, improving the efficiency of value functions learning (Mcgovern et al., 1999; Sutton et al., 1999b); (iii) enable the decision-making to take place at a higher level, (possibly) using existing controllers to execute behaviors (e.g., a navigation stack) without learning end-to-end decisions (e.g., control motor velocities). Nonetheless, limited attention has been devoted to this area of research (Xiao et al., 2020a;b), motivating the need for principled and scalable approaches. Value factorization is a natural candidate for achieving scalable asynchronous MARL, but such solutions have yet to be investigated.

This paper introduces value factorization for the macro-action framework. We formulate the *Individual Global Max (IGM)* principle for action and advantage value functions based on macro-actions.

We show both principles generalize the primitive case, representing a broader class of functions. This formalization is not trivial since it requires considering the agents' terminated and ongoing macro-actions at the joint and individual levels. We also bridge the gap between asynchronous MARL and existing primitive value factorization, introducing *Asynchronous Value Factorization* (AVF) algorithms. Core to AVF is an asynchronous (*macro-*)state collection process, which stores extra information conditioned on agents' macro-action selection. Our ablation studies show the pivotal importance of this mechanism, as algorithms trained without the macro-state fail to learn even simple cooperative behaviors. AVF enables factorization architectures to employ macro-actions, creating scalable asynchronous MARL approaches.

We evaluate the proposed methods in various problems of increasing complexity, *using standard benchmark tasks in the macro-action literature* (Xiao et al., 2020b). These problems have an increasing number of agents with multiple cooperative behaviors to learn, sub-tasks to complete, and severe partial observability. Crucially, primitive factorization and existing macro-action baselines fail to cope with the complexity of these scenarios. In contrast, AVF methods successfully learn asynchronous decentralized policies in most tasks, allowing us to achieve significantly higher payoffs and learn good decentralized behaviors where the original factorization methods fail. To our knowledge, this is the first formalization of IGM for macro-actions and value factorization macro-action-based algorithms. Our theory and approaches show impressive performance and lay the groundwork for future macro-action-based value factorization methods.

## 2 PRELIMINARIES

This section introduces two common formalizations for fully cooperative multi-agent tasks with primitive and macro-actions, which share most of the notation.[1]

Tasks with *primitive actions* are modeled as *Decentralized Partially Observable Markov Decision Processes* (Dec-POMDPs) (Oliehoek & Amato, 2016), represented as tuple $\langle \mathcal{N}, \mathcal{S}, \mathcal{U}, T_s, r, \mathcal{O}, T_\mathcal{O}, \gamma \rangle$: $\mathcal{N}$ is a finite set of agents; $\mathcal{S}$ is a finite set of states; $\mathcal{U} \equiv \langle U^i \rangle_{i \in \mathcal{N}}$ and $\mathcal{O} \equiv \langle O^i \rangle_{i \in \mathcal{N}}$ are the finite sets of primitive joint actions and observations, while $U^i, O^i$ are the individual ones. At each step, every agent $i$ chooses an action, forming the joint action $\boldsymbol{u} \equiv \langle u^i \in U^i \rangle_{i \in \mathcal{N}}$. After performing $\boldsymbol{u}$, the environment transitions from a state $s$ to a new $s'$, following a transition probability function $T_s : \mathcal{S} \times \mathcal{U} \times \mathcal{S} \rightarrow [0, 1]$ (defined as $T_s(s, \boldsymbol{u}, s') = Pr(s'|s, \boldsymbol{u})$), and returning a joint reward $r : \mathcal{S} \times \mathcal{U} \rightarrow \mathbb{R}$ for being in state $s \in \mathcal{S}$ and taking actions $\boldsymbol{u} \in \mathcal{U}$. In a partially-observable setting, agents receive an observation $\boldsymbol{o} \equiv \langle o^i \rangle_{i \in \mathcal{N}} \in \mathcal{O}$ according to an observation probability function $T_\mathcal{O} : \mathcal{O} \times \mathcal{U} \times \mathcal{S} \rightarrow [0, 1]$ (defined as $T_\mathcal{O}(\boldsymbol{o}, \boldsymbol{u}, s') = P(\boldsymbol{o}|s', \boldsymbol{u})$), and each agent maintains a policy $\pi_i(u_i|h_i)$—a mapping from local action-observation histories $h_i = (o_0^i, u_0^i, o_1^i, , \ldots, u_t^i) \in H^i$ to actions. A local history represents the actions taken and observations received at each step (up to step $t$). In finite-horizon Dec-POMDPs, the objective is to find a joint policy $\pi(\boldsymbol{u}|\boldsymbol{h})$ that maximizes the expected discounted return from a state $s_0$: $V^\pi(s_0) = \mathbb{E}_\pi \left[ \Sigma_{t=0}^{z-1} \gamma^t r_{t+1} \right]$, where $\gamma \in [0, 1)$ is a discount factor, $z$ is the problem horizon, and $\boldsymbol{h} = \langle \boldsymbol{o}_0, \boldsymbol{u}_0, \ldots, \boldsymbol{u}_t \rangle \in \boldsymbol{H}$ is the joint action-observation history.

### 2.1 VALUE FACTORIZATION

Value factorization methods learn a centralized action-value function that is factored over agent utilities and depend only on local histories for action selection. Formally, a general centralized action-value function for a policy $\pi$ can be defined as $Q^\pi(\boldsymbol{h}, \boldsymbol{u}) = \mathbb{E}_\pi [\Sigma_{t=0}^\infty \gamma^t r_{t+1} | \boldsymbol{h}, \boldsymbol{u}]$.

**Additive (VDN)** factors the joint action-value as a sum of per-agent utilities (Sunehag et al., 2018): $Q(\boldsymbol{h}, \boldsymbol{u}) = \sum_{i=1}^{|\mathcal{N}|} Q_i(h^i, u^i)$, which has good performance in many cases but can only represent a limited set of joint $Q$-functions.

**Monotonic (QMIX)** uses a non-linear monotonic mixer to combine utilities (Rashid et al., 2018):

$$\frac{\partial Q(\boldsymbol{h}, \boldsymbol{u})}{\partial Q_i(h^i, u^i)} \geq 0, \forall i \in \mathcal{N} \tag{1}$$

---

[1]For clarity, we refer to joint sets with calligraphic letters; individual ones are italic uppercase with superscript agents' indexes; joint actions, observations, and histories are lowercase bold; individual ones are lowercase. The macro-action equivalents are marked with a ˆ.

The mixer network takes extra state information as input to better factor $Q(\boldsymbol{h}, \boldsymbol{u})$ and uses positive weights to enforce the monotonicity constraint. QMIX can represent a wider range of $Q$-functions than VDN but is still limited to functions that can be factored into a non-linear monotonic combination of the agents' individual utilities.

These factorization algorithms are effective CTDE methods, maintaining consistent decentralized and joint decision-making to satisfy the IGM principle (Son et al., 2019) (Eq. 2). This is particularly important for scalability as it enables tractable joint action selection by deriving the joint greedy action from each agent's local utility. Specifically, the argmax over the joint value function is the same as argmaxing over each local utility:

$$\arg\max_{\boldsymbol{u} \in \mathcal{U}} Q(\boldsymbol{h}, \boldsymbol{u}) = \Big( \arg\max_{u^1 \in U^1} Q_1(h^1, u^1), \ \ldots, \arg\max_{u^n \in U^n} Q_n(h^n, u^n) \Big), \forall\, \boldsymbol{h} \in \boldsymbol{H} \tag{2}$$

**Advantage-based (QPLEX)** uses a decomposition of $Q$-functions to form an equivalent advantage-based IGM (*Adv-IGM*) that requires advantage values to be non-positive. $Q$-functions can be decomposed as the sum of history value and advantage functions as follows: (Wang et al., 2016):

$$Q(\boldsymbol{h}, \boldsymbol{u}) = V(\boldsymbol{h}) + A(\boldsymbol{h}, \boldsymbol{u}) \tag{3}$$

and QPLEX decomposed learned local $Q_i(h^i, u^i)$ into history and advantage utilities as follows:[2]

$$V(h^i) = \max_{u^i} Q(h^i, u^i) \quad A(h^i, u^i) = Q(h^i, u^i) - V(h^i) \quad \forall i \in \mathcal{N} \tag{4}$$

Such utilities are fed into a transformation module to condition the values on extra state information. After the transformation, QPLEX factorizes the joint action-value function as Eq. 3, using an attention module to enhance credit assignment and fully represents the functions satisfying IGM (Yang et al., 2020). Crucially, QPLEX's authors show the Adv-IGM can be satisfied by decomposing utilities as Eq. 4 (which limits the range of advantage utilities to be $\leq 0$), which allows for avoiding structural constraints in the mixer (Rashid et al., 2018; Sunehag et al., 2018). While the non-negativity of the advantage is an intrinsic property of the optimal function, Wang et al. (2016) showed how learning an arbitrary advantage function separately (i.e., without a fixed, non-negative decomposition as in Eq. 4), lead to significantly higher practical performance.

## 2.2 LEARNING MACRO-ACTION-BASED POLICIES

*Macro-Action Dec-POMDPs* (MacDec-POMDPs) (Amato et al., 2019) extend Dec-POMDPs to include durative actions, augmenting the Dec-POMDP tuple with $\langle \mathcal{M}, \hat{\mathcal{O}}, T_{\hat{o}^i \in \mathcal{N}} \rangle$: where $\mathcal{M} \equiv \langle M^i \rangle_{i \in \mathcal{N}}$ and $\hat{\mathcal{O}} \equiv \langle \hat{O}^i \rangle_{i \in \mathcal{N}}$ are the set of joint macro-actions and macro-observations. Similar to the primitive case, we define joint macro-action-macro-observation histories (or *macro-histories*) $\hat{\boldsymbol{h}}_t \in \hat{\boldsymbol{H}}$ and local ones $\hat{h}^i_t \in \hat{H}^i$. Macro-actions are based on the *options* framework (Sutton et al., 1999a); an agent's $i$ macro-action $m^i$ is defined as a tuple $\langle I_{m^i}, \pi_{m^i}, \beta_{m^i} \rangle$: $I_{m^i} \subset \hat{H}^i$ is the initiation set; $\pi_{m^i}(\cdot|h^i)$ is the macro-action low-level policy; $\beta_{m^i} : H^i \to [0, 1]$ is the termination condition.[3] The different histories allow the agents to maintain the necessary information locally to know how to execute and terminate $m^i$. During decentralized execution, agents independently select a macro-action that forms the joint one $\boldsymbol{m} = \langle m^i \rangle_{i \in \mathcal{N}}$, and maintain a high-level policy $\pi_{M^i}(m^i|\hat{h}^i)$. At each step of $m^i$'s low-level policy, the agent independently accumulates the joint reward. Upon terminating its macro-action, an agent $i$ receives a macro-observation $\hat{o}^i \in \hat{O}^i$ according to a macro-observation probability function $T_{\hat{o}^i} : O^i \times M^i \times S \to [0, 1]$, defined as $T_{\hat{o}^i}(\hat{o}^i, m^i, s') = Pr(\hat{o}^i|m^i, s')$, and resets the reward accumulation for the next macro-action. The aim is to find a joint high-level policy $\pi_{\mathcal{M}}(\boldsymbol{m}|\hat{\boldsymbol{h}})$ that maximizes the expected discounted return.

**Macro-action baselines.** Fully centralized and decentralized macro-action methods have been recently proposed (Xiao et al., 2020a;b). In Cen-MADDRQN, a centralized agent maintains a joint macro-history $\hat{\boldsymbol{h}}$ and accumulates a joint reward $r(s, \boldsymbol{m}, \boldsymbol{\tau}) = \Sigma_{t=t_{\boldsymbol{m}}}^{t_{\boldsymbol{m}}+\boldsymbol{\tau}-1} \gamma^{t-t_{\boldsymbol{m}}} r_t$ where $t_{\boldsymbol{m}}$ is the starting time-step of $\boldsymbol{m}$, and $t_{\boldsymbol{m}} + \boldsymbol{\tau} - 1$ marks its termination step when *any* agent finishes its macro-action. Hence, $\boldsymbol{\tau}$ is the number of time steps between any two macro-action terminations.

---

[2]Hence, QPLEX does not learn $V_i(h^i)$, $A_i(h^i, u^i)$ in the agents' networks as in the original dueling architecture (Wang et al., 2016), which could improve performance and sample efficiency.

[3]While we consider a deterministic termination, our results can be trivially extended to a probabilistic one.

A memory buffer $\mathcal{D}$ is used to store joint transition tuples $\langle \hat{\boldsymbol{o}}, \boldsymbol{m}, \boldsymbol{m}^-, \hat{\boldsymbol{o}}', r \rangle$. At each training iteration, the centralized agent samples a mini-batch of sequential experiences from $\mathcal{D}$ and filters out the tuples where all the macro-actions are still executing. Hence, it updates the centralized macro-action-value function by minimizing the following loss:

$$\mathbb{E}_{\langle \hat{\boldsymbol{o}}, \boldsymbol{m}, \boldsymbol{m}^-, \hat{\boldsymbol{o}}', r \rangle \sim \mathcal{D}} \left[ \left( r + \gamma^{\boldsymbol{\tau}} Q' \left( \hat{\boldsymbol{h}}', \arg\max_{\boldsymbol{m}'} Q(\hat{\boldsymbol{h}}', \boldsymbol{m}' | \boldsymbol{m}^-) \right) - Q(\hat{\boldsymbol{h}}, \boldsymbol{m}) \right)^2 \right] \tag{5}$$

where $\boldsymbol{m}^- = \{ m^i \in M^i \mid \beta_{m^i \sim \pi_{M^i}(\cdot | \hat{h}^i)} = 0, \ \forall i \in \mathcal{N} \}$ is the joint macro-action for agents whose actions will continue at the next step, and $Q'$ is a target action-value estimator (van Hasselt et al., 2016). The *conditional prediction* is crucial for a correct estimation as only a few agents typically switch to a new macro-action at the next step (Xiao et al., 2020a). Dec-MADDRQN works similarly to Cen-MADDRQN but learns each agent's Q-function in a decentralized manner. Recently, Policy Gradient (PG) (actor-critic) macro-action algorithms have been proposed (Xiao et al., 2022). However, PG methods can be less sample efficient than value-based ones (Sutton & Barto, 2018). For this reason, we focus on value-based factorization as in recent literature (Yang et al., 2020; Son et al., 2019), noting our AVF can be potentially applied to the PG's critic component.

## 3 METHODS

We first introduce the IGM principles for the asynchronous framework, proving they generalize the primitive cases. Then, we extend previous factorization methods to use macro-actions by proposing their AVF versions and investigating the use of extra asynchronous information in the mixers.

### 3.1 MACRO-ACTION-BASED IGM

To achieve principled asynchronous factorization, we must ensure the consistency of greedy macro-action selection in joint and local macro-action-value functions. This is more challenging than the primitive case as macro-actions typically last for several steps. For this reason, we must define the spaces of macro-actions that have yet to terminate under the current history as well as the ones that have terminated. For the joint case, we will leverage the conditional prediction of Eq. 5 that only applies the $\arg\max$ operator on agents that have to sample a new durative action, while maintaining the same set of ongoing macro-actions. On the other hand, in the decentralized case, only the agents with a terminated macro-action select a new one based on local information. Broadly speaking, in most cases we have to enforce the macro-action selection consistency on only a subset of the agents.

**Definition 3.1** (Mac-IGM). *Given a joint macro-history $\hat{\boldsymbol{h}} \in \hat{\boldsymbol{H}}$, we define the set of macro-action spaces $M^i$ where agent's $i$ macro-action $m^i$ has terminated under local macro-history $\hat{h}^i \in \hat{\boldsymbol{h}}$ as:*

$$\textit{(Terminated macro-action spaces)} \quad \mathcal{M}^+ = \{ M^i \in \mathcal{M} \mid \beta_{m^i \sim \pi_{M^i}(\cdot | \hat{h}^i)} = 1, \forall i \in \mathcal{N} \} \tag{6}$$

*Then, for a joint macro-action-value function $Q : \hat{\boldsymbol{H}} \times \mathcal{M} \mapsto \mathbb{R}^{|\mathcal{M}|}$, if per-agent macro-action-value functions $\langle Q_i : \hat{H}^i \times M^i \mapsto \mathbb{R}^{|M^i|} \rangle_{i \in \mathcal{N}}$ exist such that:*

$$\arg\max_{\boldsymbol{m} \in \mathcal{M}} Q(\hat{\boldsymbol{h}}, \boldsymbol{m} \mid \boldsymbol{m}^-) = \begin{cases} \arg\max_{m^i \in M^i} Q_i(\hat{h}^i, m^i) & \textit{if } M^i \in \mathcal{M}^+ \\ m^i \in \boldsymbol{m}^- & \textit{otherwise} \end{cases} , \quad \forall i \in \mathcal{N} \tag{7}$$

*then, we say $\langle Q_i(\hat{h}^i, m^i) \rangle_{i \in \mathcal{N}}$ satisfies Mac-IGM for $Q(\hat{\boldsymbol{h}}, \boldsymbol{m} \mid \boldsymbol{m}^-)$.*

Definition 3.1 ensures the greedy action selection is the same for both the centralized and decentralized action selection processes only for terminated macro-actions. We can consider a Dec-POMDP to be a degenerate form of a MacDec-POMDP where the macro-actions are primitive actions that terminate after one step. We can also include the primitive actions in the macro-action set of each agent: $U^i \subset M^i$, $\forall i \in \mathcal{N}$ (Amato et al., 2014). It follows that Mac-IGM represents a broader class of functions over the primitive IGM. We provide formal proof of such a claim in Appendix A.

**Proposition 3.2.** *Denoting with*

$$F^{IGM} = \left\{ \left( Q^{IGM} : \boldsymbol{H} \times \mathcal{U} \to \mathbb{R}^{|\mathcal{U}|}, \left\langle Q_i^{IGM} : H^i \times U^i \to \mathbb{R}^{|U^i|} \right\rangle_{i \in \mathcal{N}} \right) \mid \textit{Eq. 2 holds} \right\} \tag{8}$$

$$F^{Mac\text{-}IGM} = \left\{ \left( Q^{Mac\text{-}IGM} : \hat{\boldsymbol{H}} \times \mathcal{M} \to \mathbb{R}^{|\mathcal{M}|}, \left\langle Q_i^{Mac\text{-}IGM} : \hat{H}^i \times M^i \to \mathbb{R}^{|M^i|} \right\rangle_{i \in \mathcal{N}} \right) \mid \textit{Eq. 7 holds} \right\} \tag{9}$$

*the class of functions satisfying IGM and Mac-IGM respectively, then:*

$$F^{IGM} \subset F^{Mac\text{-}IGM} \tag{10}$$

Moreover, to design the asynchronous QPLEX algorithm (i.e., AVF-QPLEX), we define the MacAdv-IGM principle that transfers the IGM onto macro-action-based advantage functions.

**Definition 3.3** (MacAdv-IGM). *Given a joint macro-history $\hat{h} \in \hat{H}$ and $\mathcal{M}^+$ (Eq. 6), for a joint macro-action-value function $Q : \hat{H} \times \mathcal{M} \mapsto \mathbb{R}^{|\mathcal{M}|}$ defined as (3), if per-agent macro-action-value functions $\langle Q_i : \hat{H}^i \times M^i \to \mathbb{R}^{|M^i|} \rangle_{i \in \mathcal{N}}$ defined as $Q_i(\hat{h}^i, m^i) = V_i(\hat{h}^i) + A_i(\hat{h}^i, m^i)$ exist such that:*

$$\arg\max_{\boldsymbol{m} \in \mathcal{M}} A(\hat{\boldsymbol{h}}, \boldsymbol{m} \mid \boldsymbol{m}^-) = \begin{cases} \arg\max_{m^i \in M^i} A_i(\hat{h}^i, m^i) & \text{if } M^i \in \mathcal{M}^+ \\ m^i \in \boldsymbol{m}^- & \text{otherwise} \end{cases} , \quad \forall i \in \mathcal{N} \tag{11}$$

*then, we say $\langle Q_i(\hat{h}^i, m^i) \rangle_{i \in \mathcal{N}}$ satisfies MacAdv-IGM for $Q(\hat{\boldsymbol{h}}, \boldsymbol{m} \mid \boldsymbol{m}^-)$.*

Defining MacAdv-IGM is also not trivial due to the agents' terminated and ongoing macro-actions at the joint and individual levels. In addition, our definition is more general than the primitive advantage-based IGM (Section 2), since it does require advantage values to be non-positive nor the decomposition of Eq. 4. Nonetheless, it remains an equivalent transformation over the Mac-IGM as shown below. We provide formal proof of such a claim in Appendix A.

**Proposition 3.4.** *The consistency requirement of MacAdv-IGM in Eq. 11 is equivalent to the Mac-IGM one in Eq. 7. Hence, denoting with*

$$F^{MacAdv\text{-}IGM} = \left\{ \left( Q^{MacAdv\text{-}IGM} : \hat{\boldsymbol{H}} \times \mathcal{M} \to \mathbb{R}^{|\mathcal{M}|}, \langle Q_i^{MacAdv\text{-}IGM} : \hat{H}^i \times M^i \to \mathbb{R}^{|M^i|} \rangle_{i \in \mathcal{N}} \right) \mid Eq\ 11\ holds \right\} \tag{12}$$

*the class of functions satisfying MacAdv-IGM, we can conclude that $F^{Mac\text{-}IGM} \equiv F^{MacAdv\text{-}IGM}$.*

Similarly to Proposition 3.2, we can also conclude that MacAdv-IGM represents a broader class of functions over the primitive Adv-IGM, and summarize the relationship between the primitive and macro-action classes of functions.

**Proposition 3.5.** *Denoting with $F^{\{Adv\text{-}IGM, MacAdv\text{-}IGM\}}$ the classes of functions satisfying Adv-IGM and MacAdv-IGM, respectively, then:*

$$F^{IGM} \equiv F^{Adv\text{-}IGM} \subseteq F^{Mac\text{-}IGM} \equiv F^{MacAdv\text{-}IGM}. \tag{13}$$

Appendix A includes all the missing proofs and further discussions.

## 3.2 ASYNCHRONOUS VALUE FACTORIZATION

The overall architecture of AVF-based algorithms is depicted in Fig. 1. On the left, we provide a high-level overview of all the primitive value factorization mixers we employ in our framework, namely VDN, QMIX, and QPLEX (Sunehag et al., 2018; Rashid et al., 2018; Wang et al., 2021). We refer to the resultant algorithms as AVF-{VDN, QMIX, QPLEX}. In more detail, the centralized network $Q_\Theta$ used at training time is composed of agents' decentralized networks $\langle Q_{\theta_i} \rangle_{i \in \mathcal{N}}$ (depicted in yellow), and a chosen mixer module $Q_\phi$ (shown in blue), where $\theta_i, \phi$ parameterize individuals and mixing networks. During execution, each agent $i$ maintains an individual local macro-history $\hat{h}^i$ to sample its macro-action $m^i$. The low-level policy associated with $m^i$ thus starts its execution

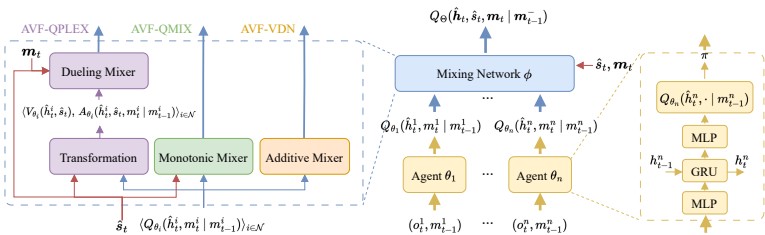

Figure 1: Overview of AVF-based architectures. We summarize the factorization methods we investigate for AVF with purple (QPLEX), green (QMIX), and orange (VDN) boxes.

at step $t_{m^i}$. The policy then continues until $\beta_{m^i}(h_{t_{m^i}+\tau-1}) = 1$ which marks its termination at step $t_{m^i} + \tau - 1$ (where $\tau$ is the length of the macro-action). Meanwhile, we accumulate the joint reward signal $r = \Sigma_{t=t_m}^{t_m+\tau-1} \gamma^{t-t_m} r_t$ that will be used to guide the centralized training procedure. Hence, upon terminating its macro-action, agent $i$ receives a new macro-observation $\hat{o}'^i$ and macro-state $\hat{s}'^i$ and updates its macro-history $\hat{h}'^i = \langle \hat{h}^i, m^i, \hat{o}'^i \rangle$. Conversely, agents that are still executing their macro-action do not update their macro-state and observation, as described in the previous section. We discuss what a macro-state is and its importance in the next section.

The agents use a centralized memory buffer $\mathcal{D}$ to store the joint transition tuple $\langle \hat{o}, \hat{s}, m, m^-, \hat{o}', \hat{s}', r \rangle$. At each training iteration, we sample a mini-batch of experiences from this buffer. As described by previous work (Xiao et al., 2020a), we filter out the experiences where none of the macro-actions have terminated yet (i.e., the intermediate execution steps of the low-level policies, where $\hat{o}'^i = \hat{o}^i$, $\hat{s}'^i = \hat{s}^i$, $\forall i \in \mathcal{N}$). We then compute the individual utilities that are fed into the (chosen) mixer, along with the joint macro-state. The mixing network employs the same architecture as the primitive case and outputs the factored joint value that drives the learning process. In summary, AVF-based algorithms are trained end-to-end to minimize Eq. 14, which resembles the fully centralized case of Eq. 5. After each training step, we update the target weights in a Polyak average fashion (Polyak & Juditsky, 1992).

$$\mathbb{E}_{\langle \hat{o}, \hat{s}, m, m^-, \hat{o}', \hat{s}', r \rangle \sim \mathcal{D}} \left[ \left( \left( r + \gamma^\tau Q_{\Theta'} \left( \hat{h}', \arg\max_{m'} Q_{\Theta} \left( \hat{h}', \hat{s}', m' \mid m^- \right) \right) - Q_{\Theta} \left( \hat{h}, \hat{s}, m \right) \right)^2 \right] \quad (14)$$

Appendix B provides the pseudocode of AVF-based algorithms, and discusses the limitations and broader impact of our framework.

**Macro-state.** *Mixing methods* that characterize recent factorization approaches, use extra information to condition the local utilities and/or the joint factored value and improve the estimation of the latter.[4] In primitive, synchronous action setups it is trivial to collect a single vector of information at each time step and use it for centralized training. However, performing asynchronous centralized training makes the process significantly harder as the utilities are computed over local histories dating back to previous steps. These should be then combined with information collected at that same step to avoid introducing noise in the factored value—we have to maintain *temporal consistency*.

Crucially, the macro-action literature does not consider factorization (Amato et al., 2019; Xiao et al., 2020b), nor the asynchronous nature that extra information has in these settings.

Consider the explanatory centralized asynchronous buffer in Fig. 2 as a practical example. For simplicity, we only discuss the case with two agents, but our discussion applies to an arbitrary number of agents. In addition, we highlight the memory compo-

Figure 2: AVF buffer; green macro-actions continue at the next step; red ones end. We consider different ways to employ extra information (blue columns).

nents that are affected by (any) macro-action termination with a dashed line at step $t_3$. Following the discussion in Section 2, we limit our example to the only samples collected when any agent terminates its macro-action (highlighted in red) since these are the only ones used to train asynchronous algorithms (Xiao et al., 2020a). In more detail, consider step $t_3$ in Fig. 2, where agent 1 terminates its macro-action $m_{t_0}^1$ that was running since step $t_0$. Upon termination, agent 1 receives a local next observation $o_{t_4}^1$ according to the real state of the environment at that moment, $s_{t_4}$ (last column in Fig. 2), and terminate the joint reward accumulation. The agent thus updates its history and samples a new macro-action $m_{t_4}^1$.[5] Both the new macro-action and the reward accumulation thus start in the *next* step $t_4$. Conversely, agent $n$ will not receive a new observation since its macro-action $m_{t_1}^n$ started at step $t_1$ and has not terminated yet.

Regarding the temporal consistency of extra state information, we investigate three different mechanisms by considering the additional blue columns depicted in Fig. 2.

---

[4]We consider the state as in factorization literature Rashid et al. (2018); Wang et al. (2021).

[5]The subscript indicates the step at which the macro-action starts; the superscript denotes the agent.

**Using the real state.** The actual state of the environment gets updated at every step, regardless of macro-action terminations. However, using this information in the centralized mixer introduces noise. In particular, the environment's state at the current step, and the state used to draw agents' observations, sample their macro-actions, and estimate utilities, typically occur at different steps as shown in the previous example. We refer to such a situation as *temporal inconsistency*. As a practical example, consider a generic mixing architecture (e.g., the ones described in Section 2) that takes as input individuals' utilities $\langle Q_i(\hat{h}^i, m^i) \rangle_{i \in \mathcal{N}}$ and the current environment's state $s$ to estimate a joint action value $Q(\hat{h}, s, m|m^-)$ satisfying Eq. 7. By applying the implicit function theorem (Krantz & Parks, 2002), such joint value can be viewed as a function of individuals' utilities. Let us discuss the temporal inconsistency problem by using the joint sample at step $t_3$:

$$Q(\hat{h}_{t_3}, s_{t_3}, m_{t_3}|m_{t_3}^-) = Q\Big(s_{t_3}, Q^1(\hat{h}_{t_0}^1, m_{t_0}^1), Q^n(\hat{h}_{t_1}^n, m_{t_1}^n)\Big) = Q\Big(Q^1(\hat{h}_{t_0}^1, s_{t_3}, m_{t_0}^1), Q^n(\hat{h}_{t_1}^n, s_{t_3}, m_{t_1}^n)\Big)$$

$$\text{where } \hat{h}_{t_3} = \langle \hat{h}_{t_0}^1, \hat{h}_{t_1}^n \rangle, \ m_{t_3} = \langle m_{t_0}^1, m_{t_1}^n \rangle, \ m_{t_3}^- = \langle m_{t_1}^n \rangle$$

Here, individual utilities are implicitly transformed using $s_{t_3}$. However, local histories and macro-actions come from the state at time $t_0, t_1$, respectively. Hence, while using future information in the mixer does not hinder Mac-IGM or MacAdv-IGM, such information does not provide useful information and only acts as noise during the joint estimation.

**Using the macro-state of each agent.** We could instead store a *macro-state* for each agent, following the same update rule of the asynchronous MARL's buffer (i.e., yellow columns in Fig. 2). Broadly speaking, each agent $i$ collects the state $\hat{s}_t^i$ of the environment at the time $t$ of selecting its macro-action $m_t^i$. The agent thus stores a transition to the next environment's state only when terminating $m_t^i$, similarly to how macro-observations are collected. We identified two ways to feed the macro-state in the mixer: (i) we can use the macro-state of the agent whose macro-action has terminated, or (ii) use a joint macro-state that comprises the macro-state of all the agents at that step. The former solution would reduce the noise in the joint estimation since at least one individual utility will be transformed using the correct macro-state. Following the previous example, we have:

$$Q(\hat{h}_{t_3}, \hat{s}_{t_0}^1, m_{t_3}|m_{t_3}^-) = Q\Big(\hat{s}_{t_0}^1, Q^1(\hat{h}_{t_0}^1, m_{t_0}^1), Q^n(\hat{h}_{t_1}^n, m_{t_1}^n)\Big) = Q\Big(Q^1(\hat{h}_{t_0}^1, \hat{s}_{t_0}^1, m_{t_0}^1), Q^n(\hat{h}_{t_1}^n, \hat{s}_{t_0}^1, m_{t_1}^n)\Big)$$

However, agent $n$, who has not terminated its macro-action, still transforms its local utility based on past (or future) information, introducing noise. In contrast, agent 1 employs the same state used for sampling its local history and macro-action and, similarly to the primitive case, this typically improves the joint estimation (Rashid et al., 2018; Wang et al., 2021; Yang et al., 2020). In contrast, the latter solution feeds the joint macro-state of all the agents (i.e., $\langle \hat{s}_{t_0}^1, \hat{s}_{t_1}^n \rangle, \rangle$ into the mixer. Despite the dimensionality of such an input, we expect the mixer to learn how to exploit the only information relevant to each individual, in order to improve the joint estimation.

Experiments in Section 4 confirm our hypothesis. Using the joint macro-state allows to learn good cooperative behaviors. In contrast, our ablation study shows that the noise introduced by using temporally inconsistent information leads to policies that fail even in our simplest tasks.

## 4 EXPERIMENTS

We investigate the scalability of AVF algorithms based on VDN (Sunehag et al., 2018), QMIX (Rashid et al., 2018), and QPLEX (Wang et al., 2021). Our experiments aim to answer the following questions: (i) *Can asynchronous factorization learn decentralized policies for complex cooperative tasks?* (ii) *Does AVF improve performance over Dec-MADDRQN (Xiao et al., 2020a)?*[6] (iii) *Are our claims regarding the noise induced by using the real state of the environments correct?* We also perform additional comparisons with primitive action factorization methods to show that primitive baselines fail at solving our complex tasks. All the data are collected on Xeon E5-2650 CPU nodes with 64GB of RAM, using the hyper-parameters in Appendix F. Considering the importance of having statistically significant results (Colas et al., 2019), we report the average return smoothed over the last ten episodes of 20 runs per method, with shaded regions representing the standard error. Finally, Appendix C also discusses the environmental impact and the solution we adopted to offset the estimated CO2 emissions of our experiments.

---

[6]We compare against Dec-MADDRQN as it also considers decentralized execution.

We use standard benchmark environments in macro-action literature (Xiao et al., 2020a;b), depicted in Fig. 3: (i) *BoxPushing* (BP). The goal is to move the big box to the goal. An agent can push the small box, but the big one requires both agents to push it simultaneously. Agents only observe the state of the cell in front of them, making high-dimensional grids significantly hard. We consider three variations BP-{10, 20, 30}, where the number indi-

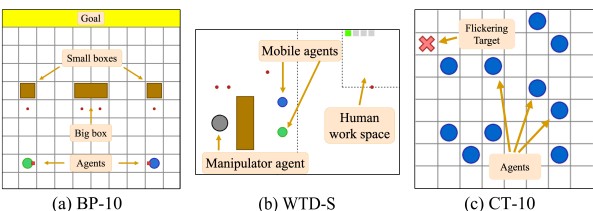

(a) BP-10          (b) WTD-S          (c) CT-10

Figure 3: Overview of BP (a), WTD (b), and CT (c) tasks. We consider multiple settings that differ by complexity (e.g., size, task prioritization) and number of agents.

cates the size of the grid. (ii) *Warehouse Tool Delivery* (WTD). A continuous space scenario with multiple workshops, each with a human worker assembling an item. Four work phases are required to complete the item, and each phase requires a tool. The manipulator has to search for the right tool and hand it to the mobile robots, which then have to deliver it to the human worker. Agents must learn the correct tools for each phase while observing the system's state only when close to a workstation. We consider three variants. WTD-S: one working human that works at a fixed speed and two mobile robots. WTD-D: two working humans with a faster first work phase and two mobile robots. WTD-T: three working humans with slower work phases and two mobile robots. WTD-T1: three working humans with different speeds and three mobile robots. (iii) *Capture Target* (CT). A group of agents has to capture a randomly moving target simultaneously. When successful, agents get a reward of 1. Agents observe their position and the target's location with a flickering probability of 0.3. We significantly increased the complexity of the original CT by considering 10 agents and 1 target (CT-10). Appendix D contains an exhaustive description of the tasks.

**AVF performance.** Fig. 4 shows the results of our evaluation. As discussed in Section 1, Dec-MADDRQN performs well in the small BP-10 task but fails at learning the optimal joint policy in larger grids. In the challenging WTD tasks, it obtains the lowest return across all the problem variations. Regarding AVF algorithms, AVF-VDN has comparable results to Dec-MADDRQN in BP domains but learns behaviors with higher payoffs in WTD-{D, T, T1}. Despite improving over the fully decentralized case, the limited class of functions that VDN can represent (Sunehag et al., 2018) leads AVF-VDN to obtain the lowest performance over the proposed asynchronous factorization methods. Similarly, the monotonicity constraint prevents AVF-QMIX from correctly factorizing value functions for which an agent's best action depends on a simultaneous action of another agent (Rashid et al., 2018). Such behavior is pivotal in tightly cooperative settings, as the optimal policy consists of agents performing a specific action simultaneously. Nonetheless, AVF-QMIX's mixer relies on our joint macro-state information and learns behaviors with higher payoffs than Dec-MADDRQN and AVF-VDN in most environments. Finally, AVF-QPLEX addresses the limited expressiveness of previous approaches, achieving better performance in all the domains.

To further confirm the superior performance of AVF-QPLEX, we performed additional experiments in CT-10. For an exhaustive comparison, we include two more baselines, the fully centralized value-based Cen-MADDRQN Xiao et al. (2020a), and the CTDE policy gradient Mac-IAICC Xiao

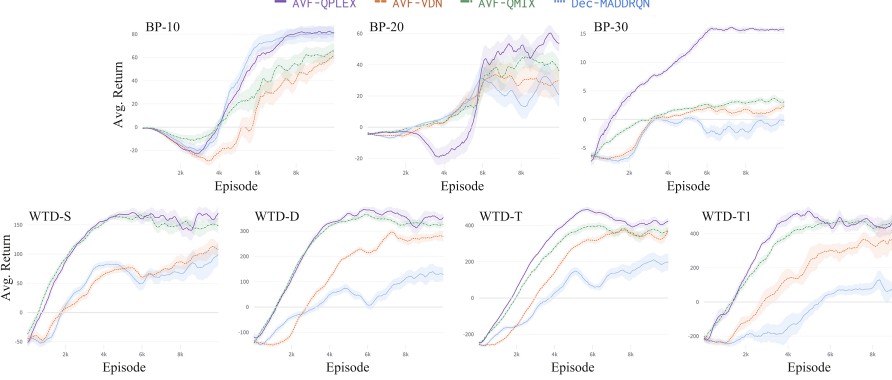

Figure 4: Comparison of Asynchronous Value Factorization algorithms and Dec-MADDRQN (Xiao et al., 2020a) in BoxPushing (top) and Warehouse Tool Delivery (bottom) tasks.

Table 1: Average return of AVF algorithms and the baselines in CT-10.

| Dec-MADDRQN | Cen-MADDRQN | AVF-VDN | AVF-QMIX | AVF-QPLEX | Mac-IAICC |
|---|---|---|---|---|---|
| $0.05 \pm 0.02$ | $0.35 \pm 0.09$ | $0.13 \pm 0.05$ | $0.26 \pm 0.08$ | $0.61 \pm 0.09$ | $0.29 \pm 0.06$ |

et al. (2022).[7] Results in Table 1 further confirm our intuitions as Dec-MADDRQN fails at solving the task, while Mac-IAICC and Cen-MADDRQN achieve higher performance than most AVF baselines. Nonetheless, only our AVF-QPLEX is able to solve the problem in most of the runs.

**Ablation study.** Fig. 5 shows the key role of the joint macro-state, using BP-10 and WTD-S as representative examples. We consider AVF-{QMIX, QPLEX}, feeding into the mixer: (i) the real state of the environment (i.e., blue rightmost column of Fig. 2), or (ii) the macro-state of the only agents which terminated their macro-action. Hence, the mixer introduces noise in the

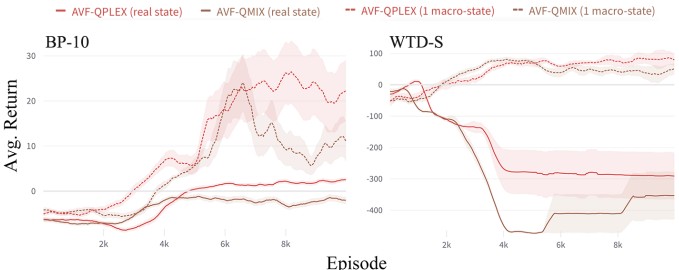

Figure 5: Results using the real state or a single macro-state for the mixers in BP-10 and WTD-S.

joint estimation. Both methods achieve negative payoffs, in contrast to the return of AVF methods that use the joint macro-state (Fig. 4). These results confirm our intuition of Section 3.2 and the importance of temporal consistency, as noisy joint action-value estimations caused by temporally uncorrelated states hinder the learning of a good performing, joint asynchronous policy.

**Primitive factorization and PG results.** Given the assumptions of MacDec-POMDPs, comparing over synchronous, primitive action methods does not provide a fair comparison. Nonetheless, we performed experiments with the primitive VDN, QMIX, and QPLEX in the primitive action versions of BP-10, WTD-S, CT-10, to show the complexity of the employed benchmarks. Appendix D.1 presents the primitive tasks and reports the average return obtained by primitive methods. Overall, VDN and QMIX fail at learning a policy that solves the tasks, while QPLEX is only capable of learning sub-optimal behaviors in all the domains.

For an exhaustive comparison, Appendix E shows the performance of the authors' implementations of Mac-IAICC (Xiao et al., 2022), and Cen-MADDRQN Xiao et al. (2020a) in the other two most complex setups, BP-30 and WTD-T1. Overall, Mac-IAICC may eventually converge to the same value as our AVF approaches but remains much less sample-efficient and more computationally demanding due to using a centralized critic for each agent. In contrast, Dec-MADDRQN has higher performance in BP-30 due to the limited size of the joint observation, but performs worse than AVF-QPLEX in WTD-T1 due to the higher complexity and dimensionality of the task.

## 5 CONCLUSION

This paper introduces Value Factorization for Asynchronous Multi-Agent Reinforcement Learning to design scalable macro-action algorithms. To this end, we proposed the IGM principle for macro-actions, ensuring consistency between centralized and decentralized greedy action selection. In addition, we showed the proposed Mac-IGM and MacAdv-IGM generalize the primitive case and represent a wider class of functions. We also introduced AVF algorithms that leverage asynchronous decision-making and value factorization. Our approach relies on a joint macro-state to maintain temporal consistency in local agents' state information, allowing the use of existing factorization architectures. Crucially, the proposed AVF framework can be applied with arbitrary mixing strategies. Extensive evaluations of complex coordination tasks showed that AVF approaches outperformed primitive factorization methods and existing macro-action baselines. Overall, our methods successfully learn asynchronous decentralized policies for challenging tasks where primitive factorization methods perform poorly. These methods show how more general action representations can be used in MARL to scale to large domains while learning high-quality solutions.

---

[7]Comparing over a PG method is unfair, due to the well-known sample inefficiency compared to value-based approaches. However, Mac-IAICC is the state-of-the-art CTDE method for macro-actions.

## 6 ETHICS AND REPRODUCIBILITY STATEMENTS

In this section, we briefly discuss two paragraphs regarding the ethics and reproducibility statements as detailed in the ICLR author guide.

**Ethics Statement.** We acknowledge our work does not contain any source of ethics concern as described in the ICLR Code of Ethics. In particular, we do not perform studies involving human subjects, nor release data sets or potentially harmful insights, methodologies, and applications. We also don't have conflicts of interest and sponsorship, discrimination/bias/fairness concerns, privacy and security issues, legal compliance, and research integrity issues.

**Reproducibility Statement.** We put significant effort into ensuring reproducibility. While we intend to release source code upon publication of our work, we included exhaustive details in multiple parts of the paper to help with reproducibility. In particular, we clearly discuss the underlying theoretical framework of Asynchronous MARL in Section 2.2. Given the strong link with the primitive case, we unified all the notation, where possible, to improve clarity. Section 2 also discusses in detail all the methodologies on which we build AVF. Similarly, Section 3 first introduces all our theoretical results, along with the related assumptions. Complete proofs for these are provided in Appendix A, and Section 4 further confirm our claims with extensive experiments in standard benchmark environments for Asynchronous MARL. Moreover, Section 4 clearly details our data collection process and how results are visualized. Finally, we included a detailed discussion about the limitations and broader impact of our work in Appendix C.

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

APPENDICES

## A    REPRESENTATIONAL COMPLEXITY OF MAC-IGM AND MACADV-IGM

As discussed by VDN and QMIX (Sunehag et al., 2018; Rashid et al., 2018), common value factorization approaches cannot guarantee representing their respective classes of true value functions in a Dec-POMDP. The same limitation holds in MacDec-POMDPs; agents' observations do not represent the full state in partially observable settings. Similarly, per-agent value function ordering can (potentially) be wrong in a macro-action context. Formally, given an agent $i$ at a time step $t$ it could happen that:

$$Q_i(\hat{h}^i, m^i) > Q_i(\hat{h}^i, m'^i) \text{ when } Q(s, (\boldsymbol{m}^{-i}, m^i)) < Q\left(s, (\boldsymbol{m}^{-i}, m'^i)\right)$$

where $\boldsymbol{m}^{-a}$ is the joint action of all the agents excluding $i$. However, there are several ways to alleviate such an issue. First, it is possible to condition per-agent action values (or state and advantage values) with state information during offline training as in QMIX (Rashid et al., 2018), QPLEX (Wang et al., 2021). Moreover, if can not assume that $(\hat{\boldsymbol{h}}, \boldsymbol{m})$ (i.e., the joint macro-history and action) is sufficient to fully model $Q(s, \boldsymbol{m})$ (which is a common assumption in prior factorization approaches), we can potentially store additional history-related information in recurrent layers (Sunehag et al., 2018).

### A.1    REPRESENTATIONAL EXPRESSIVENESS OF AVF ALGORITHMS

The proposed AVF framework does not change the architectural design of the chosen factorization method. Hence, the algorithms investigated in Section 4, namely AVF-{VDN, QMIX, QPLEX}, maintain the same considerations of the original factorization methods in terms of representational expressiveness.

In particular, AVF-VDN can factorize arbitrary joint macro-action value functions that can be additively decomposed into individual utilities. AVF-QMIX extends the family of factorizable functions to non-linear monotonic combinations. Finally, AVF-QPLEX does not involve architectural constraints and is capable of achieving the entire class of functions satisfying the underlying IGM.

#### A.1.1    OMITTED PROOFS IN SECTION 3

**Proposition 3.2.** *Denoting with*

$$F^{IGM} = \left\{ \left( Q^{IGM} : \boldsymbol{H} \times \mathcal{U} \to \mathbb{R}^{|\mathcal{U}|}, \left\langle Q_i^{IGM} : H^i \times U^i \to \mathbb{R}^{|U^i|} \right\rangle_{i \in \mathcal{N}} \right) \mid Eq. \ 2 \ holds \right\} \tag{15}$$

$$F^{Mac\text{-}IGM} = \left\{ \left( Q^{Mac\text{-}IGM} : \hat{\boldsymbol{H}} \times \mathcal{M} \to \mathbb{R}^{|\mathcal{M}|}, \left\langle Q_i^{Mac\text{-}IGM} : \hat{H}^i \times M^i \to \mathbb{R}^{|M^i|} \right\rangle_{i \in \mathcal{N}} \right) \mid Eq. \ 7 \ holds \right\} \tag{16}$$

*the class of functions satisfying IGM and Mac-IGM respectively, then:*

$$F^{IGM} \subset F^{Mac\text{-}IGM} \tag{17}$$

*Proof.* MacDec-POMDPs extends Dec-POMDPs by replacing the primitive actions available to each agent with option-based macro-actions. However, as shown in (Amato et al., 2019), the macro-action set contains primitive actions to guarantee the same globally optimal policy:

$$U^i \subset M^i, \ \forall i \in \mathcal{N} \tag{18}$$

Meaning that $\forall i \in \mathcal{N}, \ |M_i| > |U_i|$, which implies $|\mathcal{M}| > |\mathcal{U}|$. It also follows that $\mathcal{O} \subseteq \hat{\mathcal{O}}$ as a MacDec-POMDP is, in the limit where only primitive actions are selected, equivalent to a Dec-POMDP. For these reasons, we can conclude that $|\hat{\boldsymbol{H}} \times \mathcal{M}| > |\mathbf{H} \times \mathcal{U}|$ (i.e., the domain over which primitive action-value functions are defined is smaller than the domain over which macro-action-value functions are defined). Hence, $F^{\text{IGM}} \subset F^{\text{Mac-IGM}}$. $\qquad\square$

**Proposition 3.4.** *The consistency requirement of MacAdv-IGM in Eq. 11 is equivalent to the Mac-IGM one in Eq. 7. Hence, denoting with*

$$F^{MacAdv\text{-}IGM} = \left\{ \left( Q^{MacAdv\text{-}IGM} : \hat{\boldsymbol{H}} \times \mathcal{M} \to \mathbb{R}^{|\mathcal{M}|}, \langle Q_i^{MacAdv\text{-}IGM} : \hat{H}^i \times M^i \to \mathbb{R}^{|M^i|} \rangle_{i \in \mathcal{N}} \right) \mid Eq \ 11 \ holds \right\} \tag{19}$$

*the class of functions satisfying MacAdv-IGM, we can conclude that $F^{Mac\text{-}IGM} \equiv F^{MacAdv\text{-}IGM}$.*

*Proof.* Given a joint macro-history $\hat{\boldsymbol{h}} \in \hat{\boldsymbol{H}}$ on which $\langle Q_i(\hat{h}^i, m^i) \rangle_{i \in \mathcal{N}}$ satisfies Mac-IGM for $Q(\hat{\boldsymbol{h}}, \boldsymbol{m} \mid \boldsymbol{m}^-)$, we show Eq. 11 represents the same consistency constraint as Eq. 7. By applying the dueling decomposition from (Wang et al., 2016), we know $Q(\hat{\boldsymbol{h}}, \boldsymbol{m} \mid \boldsymbol{m}^-) = V(\hat{\boldsymbol{h}}) + A(\hat{\boldsymbol{h}}, \boldsymbol{m} \mid \boldsymbol{m}^-)$, and $Q_i(\hat{h}^i, m^i) = V(\hat{h}^i) + A_i(\hat{h}^i, m^i)$, $\forall i \in \mathcal{N}$. Hence, the state-value functions defined over macro-histories do not influence the action selection process. For the joint value, we can thus conclude that:

$$\arg\max_{\boldsymbol{m} \in \mathcal{M}} Q(\hat{\boldsymbol{h}}, \boldsymbol{m} \mid \boldsymbol{m}^-) = \arg\max_{\boldsymbol{m} \in \mathcal{M}} V(\hat{\boldsymbol{h}}) + A(\hat{\boldsymbol{h}}, \boldsymbol{m} \mid \boldsymbol{m}^-) = \arg\max_{\boldsymbol{m} \in \mathcal{M}} A(\hat{\boldsymbol{h}}, \boldsymbol{m} \mid \boldsymbol{m}^-) \quad (20)$$

Similarly, for the individual values:

$$\forall i \in \mathcal{N}, \begin{cases} \arg\max_{m^i \in M^i} Q_i(\hat{h}^i, m^i) & \textit{if } M^i \in \mathcal{M}^+ \\ m^i \in \boldsymbol{m}^- & \textit{otherwise} \end{cases}$$

$$= \begin{cases} \arg\max_{m^i \in M^i} V(\hat{h}^i) + A_i(\hat{h}^i, m^i) & \textit{if } M^i \in \mathcal{M}^+ \\ m^i \in \boldsymbol{m}^- & \textit{otherwise} \end{cases} \quad (21)$$

$$= \begin{cases} \arg\max_{m^i \in M^i} A_i(\hat{h}^i, m^i) & \textit{if } M^i \in \mathcal{M}^+ \\ m^i \in \boldsymbol{m}^- & \textit{otherwise} \end{cases}$$

Broadly speaking, we know the history values act as a constant for both the joint and local estimation and do not influence the $\arg\max$ operator. By combining Eq. 20, 21, we conclude the equivalence between Eq. 7, 11. $\qquad \square$

**Proposition 3.5.** *Denoting with $F^{\{Adv\text{-}IGM, MacAdv\text{-}IGM\}}$ the classes of functions satisfying Adv-IGM and MacAdv-IGM, respectively, then:*

$$F^{IGM} \equiv F^{Adv\text{-}IGM} \subset F^{Mac\text{-}IGM} \equiv F^{MacAdv\text{-}IGM}. \quad (22)$$

*Proof.* The result naturally follows from Proposition 3.2, 3.5, and the result of (Wang et al., 2021) that showed the equivalence between the class of functions represented by the primitive IGM and Adv-IGM. In more detail, from the latter we know $F^{\text{IGM}} \equiv F^{\text{Adv-IGM}}$. Moreover, Proposition 3.2 showed us that $F^{\text{IGM}} \subset F^{\text{Mac-IGM}}$, from which follows that $F^{\text{Adv-IGM}} \subset F^{\text{Mac-IGM}}$. In addition, Proposition 3.5 showed us that $F^{\text{Mac-IGM}} \equiv F^{\text{MacAdv-IGM}}$. Combining these results, we conclude the relationship in Eq. 22. $\qquad \square$

## B  PSEUDOCODE, LIMITATIONS AND BROADER IMPACT

The general flow of our asynchronous approaches is presented in Algorithm 1.

In more detail, the centralized network $Q_\Theta$ used during the training phase is composed of agents' decentralized networks $\langle Q_{\theta_i} \rangle_{i \in \mathcal{N}}$, and the chosen mixer module $Q_\phi$. The same holds for the target centralized network typically used in value-based approaches van Hasselt et al. (2016) (line 2). At each time step, if the temporally extended action of an agent $i$ ended at the previous time-step, $i$ observes the next macro observation $\hat{o}'^i$ that is then used to sample a new macro action $m^i$ within a standard $\epsilon$-greedy policy Mnih et al. (2013). It also collects its new macro-state $\hat{s}^i$ that some of the factorization architectures will employ. The cumulate reward signal also resets to 0 (lines 6-11). After executing a step of their low-level policies $\pi_{m^i}$, we accumulate the joint reward and receive a new macro observation $\hat{o}'^i$ if $m^i$ ends (lines 12-14). A joint sample is stored in the centralized buffer $\mathcal{D}$ (line 15). At training time, we sample and squeeze a batch of trajectories from $\mathcal{D}$ as described in the next section. We then compute the per-agent action-values as in DDRQN Hausknecht & Stone (2015) (lines 16-18). Such action-values are fed into the mixer with (possibly) extra state information to compute the factored values required by the loss in Eq. 14 to perform a gradient descent step (lines 19-20). Finally, we update the target weights in a Polyak average fashion Polyak & Juditsky (1992) (line 21).

---

**Algorithm 1** Asynchronous Value Factorization

---

1: **Given:**
- Agents decentralized and target networks $\langle Q_{\theta_i, \theta_i'} \rangle_{i \in \mathcal{N}}$
- Mixer and target mixer networks $Q_\phi, Q_{\phi'}$
- Centralized memory buffer $\mathcal{D}$
- Initial macro observations and macro-states $\langle \hat{o}^i, \hat{s}^i \rangle_{i \in \mathcal{N}}$
- N° of episodes $e$, time-out $t'$, update frequency $f$, target network update coefficient $\omega$

2: Centralized $Q_{\Theta}$ and target $Q_{\Theta'}$ networks for training are composed by $(\langle Q_{\theta_i} \rangle_{i \in \mathcal{N}}, Q_\phi)$, $(\langle Q_{\theta_i'} \rangle_{i \in \mathcal{N}}, Q_{\phi'})$

3: **for** *episode* = 1 to $e$ **do**
4:    **for** $t = 0$ to $t'$ **do**
5:       **for** each agent $i$ **do**
6:          **if** $m^i$ is terminated **then**
7:             Reset cumulative reward $r$
8:             $\hat{o}^i \leftarrow \hat{o}'^i$
9:             Get the macro-state $\hat{s}^i$ from the environment
10:             $m^i \sim \epsilon$-greedy policy using $Q_{\theta_i}(\hat{h}^i, m^i)$
11:          **end if**
12:       **end for**
13:    Execute $\boldsymbol{m} = \{m^i\}_{i \in \mathcal{N}}$ in the environment
14:    Accumulate joint reward $r$
15:    $\hat{o}' \leftarrow \langle \hat{o}'^i \rangle_{i \in \mathcal{N}}$; $\forall i$, if $m^i$ does not end, $\hat{o}'^i = \hat{o}^i$
16:    Store $\langle \hat{\boldsymbol{o}}, \boldsymbol{m}, \hat{\boldsymbol{o}}', r, \hat{\boldsymbol{s}} \rangle$ into $\mathcal{D}$
17:    **if** $t \% f = 0$ **then**
18:       Sample and filter trajectories as in Section 3.2
19:       Compute per-agent action-values using $\langle Q_{\theta_i, \theta_i'}, \hat{h}^i, m^i, \hat{h}'^i \rangle_{i \in \mathcal{N}}$
20:       Factorize action-values using $Q_\phi, Q_{\phi'}, \hat{\boldsymbol{s}}$
21:       Perform a gradient descent step on $\mathcal{L}(\Theta)$ following Eq. 5 on the mixed values
22:       Update target weights $\Theta' \leftarrow \omega \Theta' + (1 - \omega) \Theta$
23:    **end if**
24:    **end for**
25: **end for**

---

## B.1 LIMITATIONS

We mainly identify three limitations in the proposed AVF framework:

- Most previous factorization approaches cannot guarantee to fully represent their respective classes of value functions in a Dec-POMDP (Sunehag et al., 2018; Rashid et al., 2018; 2020); the same limitation holds in AVF-based algorithms that maintain the same representation expressiveness of the original methods. In particular, agents' observations do not represent the true state in partially observable settings, and per-agent value function ordering can (potentially) be wrong also in a macro-action context (Rashid et al., 2018).

- AVF-{QMIX, QPLEX} methods employ extra information per agent during factorization, which could cause scalability issues when considering many agents. While such a problem does not arise in our experiments, it would be possible to train an encoder to feed into AVF's mixers a compact representation of per-agent data.

- MacDec-POMDPs assume that low-level policies for executing macro-actions are known and fixed. However, asynchronous decision-making settings are common in the real world but have been rarely studied in the previous MARL literature. For this reason, principled methods are needed for the fixed (MacDec-POMDP) case before extending them to learn macro-actions (e.g., by employing skill discovery approaches to learn macro-actions (Eysenbach et al., 2019)).

## B.2 BROADER IMPACT

Regarding the broader impact of our work, we do believe macro-actions have the potential to scale MARL into the real world. Temporally extended actions enable decision-making at a higher level and naturally represent complex real-world behavior (e.g., lifting an object). that can exploit existing robust controllers or be defined by a (human) expert, making them more explainable than other sequences of primitive actions. By extending MARL algorithms to the macro-action case, realistic multi-agent coordination problems can be solved that are orders of magnitude larger than problems solved by previous primitive MARL algorithms.

## C ENVIRONMENTAL IMPACT

Despite being significantly less computationally demanding than conventional Deep Learning approaches, training agents with MARL algorithms leads to environmental impacts due to intensive computations that (possibly) run on computer clusters for an extended time. Nonetheless, it is crucial to foster sample efficiency (i.e., reducing the training time for the agents, hence the computational resources used to train them) to reduce the environmental footprint of such learning systems. In this direction, our work considers designing macro-action methods that significantly improve the sample efficiency of the learning algorithms (i.e., the number of simulation steps required to learn a policy), as shown by previous research on the topic (Xiao et al., 2020a;b). Our experiments were conducted using a private infrastructure with a carbon efficiency of $\approx 0.275 \frac{kgCO_2eq}{kWh}$, requiring a cumulative $\approx$72 hours of computation. Total emissions are estimated to be $\approx 4.26 kgCO_2eq$ using the Machine Learning Impact calculator of which 100% were offset through Treedom.

## D DOMAIN DESCRIPTION

### D.1 BOX PUSHING (BP)

In this collaborative task, two agents have to work together to push a big box to a goal area at the top of a grid world to obtain a higher credit than pushing the small box on each own. The small box is movable with a single agent, while the big one requires two agents to push it simultaneously.

The state space consists of each agent's position and orientation, as well as the location of each box. Agents have a set of primitive actions, including *moving forward*, *turning left* or *right*, and *staying* in place. The available macro-actions are **Go-to-Small-Box(i)** and **Go-to-Big-Box** that navigates the agent to a predefined waypoint (red) under the corresponding box and terminates with a pose facing it; and a **Push** macro-action that makes the agent move forward and terminate when the robot hits the world boundary or the big box. Each agent observation is very limited in both the primitive and macro level, which is the state of the front cell: empty, teammate, boundary, small box, or big box.

The team receives a terminal reward of $+100$ for pushing the big box to the goal area or $+10$ for pushing one small box to the goal area. If any agent hits the world's boundary or pushes the big box on its own, a penalty of $-10$ is issued. An episode terminates when any box is moved to the goal area or reaches the maximum horizon, 100 time steps. In our work, we consider the variant of this task in terms of the grid world size as shown in Fig. 6.

The original work of Xiao et al. (2020a) also released a primitive action version of the BP task. In the primitive action version, each agent has four actions: move forward, turn left, turn right, and stay. The small box moves forward one grid cell when any robot faces it and executes the move *move forward* action.

### D.2 WAREHOUSE TOOL DELIVERY (WTD)

Warehouse Tool Delivery scenarios vary in the number of agents, humans, and the speed at which they work. In each scenario, the humans assemble an item with four work phases. Each phase requires several primitive time steps and a specific tool. We assume that the human already holds the tool for the first phase, and the rest must be found and delivered in a particular order by a team of robots to finish the subsequent work phases. The objective of the robot team is to assist the humans

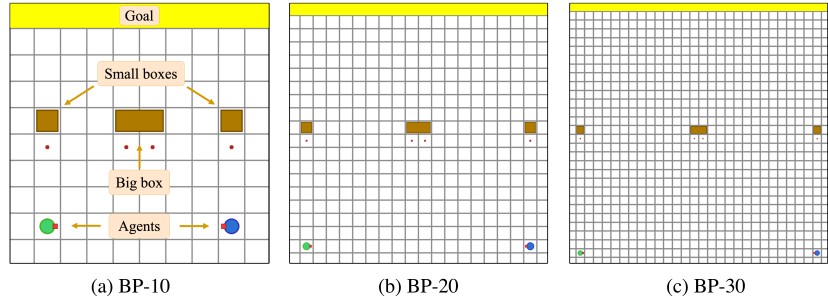

Figure 6: Overview of the considered Box Pushing task variations.

in completing their tasks as quickly as possible by finding and delivering the correct tools in the proper order and timely fashion without making the humans wait.

The environmental space is continuous, and the global state includes 1) each mobile robot's 2D position; 2) the execution status of the manipulator robot's macro-action in terms of the rest of primitive time steps to terminate; 3) the work phase of each human with its completed percentage; and 4) each tool's position.

Mobile robots have three navigation macro-actions: 1) **Go-W(i)** moves the robot to the corresponding workshop and locates at the red spot in the end; 2) **Go-TR** leads the robot to the red waypoint in the middle of the tool room; 3) **Get-Tool** navigates the robot the pre-allocated waypoint beside the manipulator and wait there, which will not terminate until either receiving a tool or waiting there for 10 time steps. Mobile robots move at a fixed velocity and are only allowed to receive tools from the manipulator rather than the human. There are three applicable macro-actions for the manipulator robot: 1) **Search-Tool(i)** takes 6 time steps to find a particular tool and place it in a staging area when there are less than two tools there; otherwise, it freezes the robot for the same amount of time.; 2) **Pass-to-M(i)** takes 4 time steps to pick up the first found tool from the staging area and pass it to a mobile robot; 3) **Wait-M** consumes 1 time step to wait for a mobile robot.

Each mobile robot is always aware of its location and the type of tool carried by itself. Meanwhile, it is also allowed to observe the number of tools in the staging area or a human's current work phase when it is at the tool room or the corresponding workshop, respectively. The macro-observation of the manipulator robot is limited to the type of tools present in the staging area and the identity of the mobile robot waiting at the adjacent waypoints.

Rewards for this domain are structured such that the team earns a reward of $+100$ when they deliver a correct tool to a human on time. However, if the delivery is delayed, an additional penalty of $-20$ is imposed. Moreover, the team incurs a penalty of $-10$ if the manipulator robot attempts to pass a tool to a mobile robot that is not adjacent, and a penalty of $-1$ happens every time step.

We consider four variations of WTD shown in Fig. 7: a) WTD-S, involves one human and two mobile robots; b) WTD-D, involves two humans and two mobile robots; c) WTD-T, involves three humans and two mobile robots. d) WTD-T1, involves three humans and three mobile robots. The human working speeds under different scenarios are listed in Table 2

Table 2: The number of time steps each human takes on each working phase in scenarios.

| Scenarios | WTD-S | WTD-D | WTD-T | WTD-T1 |
|---|---|---|---|---|
| Human-0 | $[20, 20, 20, 20]$ | $[27, 20, 20, 20]$ | $[40, 40, 40, 40]$ | $[38, 38, 38, 38]$ |
| Human-1 | N/A | $[27, 20, 20, 20]$ | $[40, 40, 40, 40]$ | $[38, 38, 38, 38]$ |
| Human-2 | N/A | N/A | $[40, 40, 40, 40]$ | $[27, 27, 27, 27]$ |

Each episode stops when all humans have obtained the correct tools for all work phases or when the maximum time steps are reached (150 for WTD-S, 150 for WTD-D, and 150 for WTD-T).

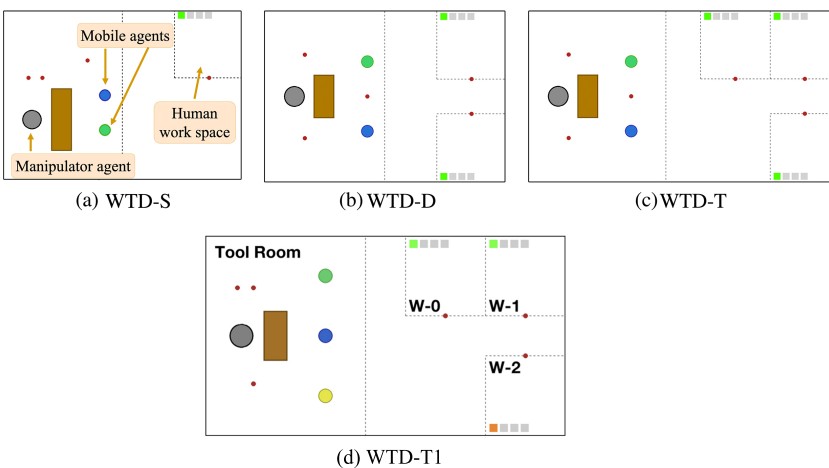

Figure 7: Overview of the considered Warehouse Tool Delivery task variations.

### D.3 CAPTURE TARGET (CT)

In this domain, there are 10 agents represented by blue circles, assigned with the task of capturing a randomly moving target indicated by a red cross (as shown in Fig. 3). Each agent's macro-observation captures the same information as its primitive one, including the agent's position (being always observable) and the target's position (being partially observable with a flickering probability of 0.3). The applicable primitive-actions include moving *up*, *down*, *left*, *right*, and *stay*. The macro-action set consists of ***Move-to-T***, directs the agent to move towards the target with an updated target position according to the latest primitive observation, and *Stay* lasts a single time step. The horizon of this task is 60 time steps, and a terminal reward of $+1$ is given only when all agents capture the target simultaneously by being in the same cell.

### D.4 RESULTS FOR THE PRIMITIVE FACTORIZATION BASELINES

Table 3 reports the performance of the primitive action baselines VDN, QMIX, QPLEX in the primitive BP-10, WTD-S, CT-10. The primitive version of these environments were obtained by constraining macro-actions to one-step action. Crucially, as discussed in Section 4, primitive baselines struggle to cope with the complexity of the tasks that are typically used in the macro-action literature, since they require a high degree of cooperation, and are characterized by significant partial observability.

Table 3: Experiments with primitive factorization VDN, QMIX, QPLEX in the primitive version of BP-10, WTD-S, CT-10.

|         | VDN             | QMIX            | QPLEX           |
|---------|-----------------|-----------------|-----------------|
| BP-10   | -4.1 $\pm$ 10.5 | -2.0 $\pm$ 9.3  | 13.4 $\pm$ 2.6  |
| WTD-S   | 32 $\pm$ 4.5    | 44 $\pm$ 6.2    | 61 $\pm$ 5.3    |
| CT-10   | 0.04 $\pm$ 0.01 | 0.09 $\pm$ 0.03 | 0.20 $\pm$ 0.04 |

## E COMPARISON WITH CEN-MADDRQN AND MAC-IAICC

Fig. 8 shows the performance of the authors' implementations of Mac-IAICC (Xiao et al., 2022), and Cen-MADDRQN Xiao et al. (2020a) in the two remaining most complex setups, BP-30 and WTD-T1. As discussed in the main paper, Mac-IAICC may eventually converge to the same value as AVF-QPLEX, but remains much less sample-efficient and more computationally demanding due to using a centralized critic for each agent. In contrast, Dec-MADDRQN has higher performance

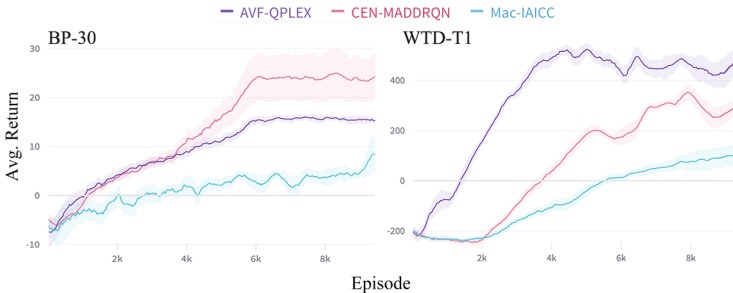

Figure 8: Comparison of our best performing algorithm, AVF-QPLEX against the fully centralized value-based Cen-MADDRQN Xiao et al. (2020a) and the CTDE policy-gradient Mac-IAICC Xiao et al. (2022) in the remaining most complex scenarios, BP-30 and WTD-T1

than AVF-QPLEX in BP-30, due to the limited size of the joint observation, but performs worse in WTD-T1 due to the higher complexity and dimensionality of the task. Overall these results confirm the merits of the proposed asynchronous value factorization framework.

## F   HYPER-PARAMETERS

Regarding the considered baselines, we employed the original authors' implementations and parameters (Sunehag et al., 2018; Rashid et al., 2018; Wang et al., 2021; Xiao et al., 2020a; 2022). Table 4 lists all the hyper-parameters considered in our initial grid search for tuning the algorithms employed in Section 4. We separate algorithm-specific parameters (e.g., for the mixer of AVF-QMIX, AVF-QPLEX) with a horizontal line at the end of the table. We tested different joint reward schemes for macro-actions (e.g., only considering the max/min values and time horizon among the agents, averaging them). Still, the original joint scheme in Section 2.2 resulted in the best performance.

Table 4: Hyper-parameters candidate for initial grid search tuning.

| | |
|---|---|
| Learning rate | 1e-3, 3e-4, 1e-4, 5e-5 |
| $\gamma$ | 0.9, 0.95, 0.99 |
| ASVB (full episodes) size | 1000, 2500, 5000 |
| Batch size | 32, 64, 128 |
| Sampling trajectory size | 10, 25, 50 |
| Polyak averaging $\omega$ | 0.995, 0.9998 |
| N° hidden layers | 2, 3 |
| Hidden layers size | 64, 128 |
| Mix embed. size | 32, 64 |
| Hypernet embed. size | 32, 64 |
| N° hypernet layers | 2 |
| N° Advantage hypernet layers | 2 |
| Advantage hypernet embed. size | 32, 64 |

Table 5 lists the hyper-parameters considered in our experiments. When a parameter differs from the BP and the WTD domains, we indicate both values with a separator. Shared parameters between all the algorithms are indicated once.

Table 5: Hyper-parameters used in our experiments.

|  | AVF-VDN | AVF-QMIX | AVF-QPLEX |
|---|---|---|---|
| Learning rate | 5e-5 — 1e-4 | 5e-5 — 1e-4 | 1e-4 — 1e-4 |
| $\gamma$ | 0.9 | 0.99 — 0.9 | 0.99 — 0.9 |
| ASCB size |  | 2500 |  |
| Batch size |  | 64 |  |
| Sampling traj. size |  | 25 |  |
| $\omega$ |  | 0.995 |  |
| N° hidden layers |  | 2 |  |
| Hidden layers size |  | 64 |  |
| Mix embed. size | - | 32 | 32 |
| Hypernet embed. size | - | 32 | - |
| N° hypernet. layers | - | 2 | - |
| N° Adv. hypernet layers | - | - | 2 |
| Adv. hypernet embed. size | - | - | 32 |

