# OpenReview forum: "Value Factorization for Asynchronous Multi-Agent Reinforcement Learning"
_ICLR.cc/2024/Conference — Submitted to ICLR 2024_

### Official Review · Reviewer_eKQs · 2023-10-14

**Soundness:** 3 good
**Presentation:** 2 fair
**Contribution:** 3 good
**Rating:** 6
**Confidence:** 4

**Summary:**

The paper introduces value-factorization for settings with temporally-extended actions (macro-actions). The Individual-Global Maximum (IGM) property is first adapted to such setting, and then different algorithms (based on those already existing in literature for the primitive-action case) are derived. Experimental results show the improvements of value-factorization is such setting over existing decentralized solutions.

**Strengths:**

The investigated setting is interesting: indeed, macro-actions are a more general and accurate way to represent real-world multi-agent problems, and can thus lead to a wider application of these. With value-factorization being one of the most prominent technique investigated in MARL recently, it seems natural to derive such an extension. Also, the way in which it is derived, as well and the whole discussion about the use of macro-states and the updates only on terminated macros is logical and sound. Most of the experimental results are clear and really show the benefits of value-factorization methods over non-factored ones.

**Weaknesses:**

While the paper is generally clear in its notation and easy to read, some passages appear to be less readable and a bit more messy. Also, some of the experimental results, although going in the right direction, are not sound enough to actually credit the proposed theoretical claims. Please see the Questions below for a more detailed explanation.

**Questions:**

- The background section is a bit rushed. For example, you mention the Q-function in Section 2.1, but you are never actually defining it formally or even explaining what that is intuitively. Although probably every reader of this paper will already know this, the point of the Background section is exactly to provide a brief recap of such basic notions, and thus I think it is important to be as accurate and comprehensive as possible here.
- Why not requiring the values not to be non-positive in your proposed Adv-IGM is a more general condition? The non-positive nature of the advantage function is not a superimposed limitation, but rather an intrinsic property of this function, and as such I do not clearly see how this can be defined as a limitation. Also, it is not entirely clear to me how you can avoid the decomposition in Equation (4) with your proposed formulation. Please argument a bit more in this respect to make your claims clearer.
- Why not comparing also to Cen-MADDRQN? It would have been interesting to compare the obtained results not only against a decentralized solution (to assess the improvements of the proposed technique w.r.t. that), but also against a centralized solution, to compare how well is your decentralized technique doing against a (more expensive) centralized one, as well as assessing how such a decentralized technique can scale better when problems become more complex.
- The comparison results of different mixers seems a bit expected: these are indeed what one would expect from the standard literature in the primitive-action setting. Indeed, QMIX is a generalization of VDN (additivity is a sub-case of monotonicity), and QPLEX in a generalization of both (as monotonicity is a sufficient but not necessary condition for IGM). What is the added value of such a comparison here? What is the additional bit of knowledge that the reader can gain from your proposed results?
- The comparison between using the true state vs. using the macro-state in AVF is interesting, but only reporting results on a single problem is not convincing enough to claim that the true state is not a good conditioning signal. Could you please add similar results on another benchmark to prove that your claim generalizes beyond the specific WTD-S setting?
- Table 2 is a bit confusing: I appreciate the idea of comparing primitive-action algorithms with their macro-action counterparts, but the proposed results are not actually clear in this sense. You propose results for BP-10 for the primitive-action algorithms, but then you move to BP-30 and WTD-T1 for your AVF-QPLEX (and the proposed macro-action PG method). How can we make a good comparison out of that? Would not it have been better to compare these on the same set of benchmarks?
- Also, results of the primitive-action algorithms and their macro-action counterparts on BP-10 does not seem to match: as you claimed in Section 3, your proposed methods should reduce to the primitive case when macro-actions have a single-step duration. So what is the difference here? Why are their performances not matching with those shown in Figure 4? If you are comparing these primitive-action algorithms on setting with a temporally-extended actions, that would be an unfair comparison. If not, however, I would expect their performance to be on-par with your AVF versions. Could you explain this a bit more in details?

---

> ### Author Response · Authors · 2023-11-13
> **Rebuttal by Authors**
>
> We appreciate Reviewer eKQs’ positive evaluation of our work and we thank the reviewer for the insightful questions. We believe our revised paper has significantly improved by addressing your doubts. In the following, we summarize the changes in the manuscript that answer your questions.
>
> - We agree the background section could appear rushed as we omitted a few details to fit the page limit. We revised it, including a definition for Q, clarified the meaning of $\tau$, and further detailed the limit of QPLEX’s decomposition (discussed in the next point). We kindly ask the reviewer to point out any additional missing details in Section 2 that we will carefully incorporate.
>
> - Regarding Adv-IGM, we agree the non-positive nature of the advantage function is an intrinsic property of it. However, considering the discussion in the dueling networks’ work [A], allowing the advantage function to assume arbitrary values typically leads to significantly higher performance. As such, Adv-IGM represents a more general condition as it will allow future factorization schemes to learn arbitrary advantage functions, inheriting the benefits of the single-agent dueling networks case. For example, the decomposition of Eq. 4 could be avoided by employing a principled version of dueling networks, where agents learn separate streams for state and advantage value functions (without any value restriction). This is an exciting direction to investigate in the future but goes beyond the scope of our paper.
>
> - Regarding Cen-MADDRQN, we performed additional experiments (each over 20 independent runs) in the most complex domains: BP-30, WTD-T1, and CT-10 (Table 1 and Appendix E). In CT-10, Cen-MADDRQN showed poor performance due to the high number of agents employed in the task. Similarly, Cen-MADDRQN also has lower performance in WTD-T1 with respect to our best-performing solution, AVF-QPLEX. Conversely, BP-30 is characterized by a few agents and a joint observation of limited size, and the fully centralized baseline obtains higher performance than AVF methods.
>
> - Regarding the results of different mixers, extending factorization methods to asynchronous MARL requires the asynchronous buffer and the use of temporal consistent joint information. The underlying mixing architectures of the primitive algorithms remain the same, and it makes sense the general results are the same in the primitive setting. The added value in our experiments is the ablation study conducted to support our claims on the importance of temporal consistency. Without using a joint macro-state, in fact, AVF methods fail at solving even the easiest macro-action tasks (BP-10 and WTD-S).
>
> - Concerning the ablation study, we performed an additional set of experiments in BP-10. In particular, the new Figure 5 shows how AVF methods based on the true state fail at solving the easiest configuration of BP and WTS. The high number of independent runs we consider in all our experiments is also a strong indicator of the significance of these results. We further enhanced our ablation study by considering AVF-{QMIX, QPLEX} employing the only macro-state of the agent which terminated its macro-action (i.e., the second mechanism discussed in Section 3.2.). As expected, the noise introduced by such a method leads to significantly lower performance than the joint macro-state employed in Fig. 5, but represents a better-performing solution than using the true state.
>
> - We agree Table 2 is confusing. We wanted to show how primitive-action methods fail in the easiest (primitive) task, BP-10, to highlight the complexity of the benchmarks employed in the macro-action literature. Conversely, when comparing with the state-of-the-art Mac-IAICC, we used the most complex scenarios to highlight the superior performance of AVF. Following our additional experiments with Cen-MADDRQN, we moved the comparison with primitive methods in Appendix D.1, and discuss the comparison with Mac-IAICC and Cen-MADDRQN in Appendix E. We believe such a revision significantly improves clarity, but we remain available to further revise our manuscript if you believe that would further improve our work.
>
> - There has been a misunderstanding on the results of the primitive-action algorithms. We tested the primitive version of VDN, QMIX, and QPLEX in the primitive version of BP-10 presented in (Xiao et al., 2020a). These primitive methods only employ 1-step moving and pushing actions, as they cannot employ the temporally extended actions of their asynchronous counterpart. We added a description of the primitive BP in Appendix D.1. As clearly stated in Section 4, we remark such a comparison is not fair since primitive methods cannot leverage the expert knowledge of low-level policies used by macro-action-based methods. Intuitively, this is why their performance is not on par with the asynchronous versions.
>
> [A] Z. Wang et al., “Dueling Network Architectures for Deep Reinforcement Learning,” ICML, 2016.

---

> > ### Comment · Reviewer_eKQs · 2023-11-20
> > **Reply to authors**
> >
> > I would like to thank the authors for their replies, these helped in clarifying some of my doubts. Here are some additional points:
> > 2. I still do not see clearly why allowing the advantage function to get also positive values may be of any help. The advantage function is defined to be negative, so if during learning it takes a positive value then that value is inherently wrong. Could you please clarify briefly how allowing the network to do this is helpful rather than harmful indeed?
> > 3. Yes, the results with different mixers are expected indeed. So again, what is the benefit of having these there (I have not objected to the ablation study indeed, which I clearly see why is useful)?
> > 5. Isn't the asynchronous task with unitary action duration exactly the corresponding primitive task? So the comparison with primitive algorithms should make sense as long as you stick to this formulation right? I still do not get what are the difference between the primitive environments/algorithms and their unitary asynchronous counterparts, and why you would eventually report results for a comparison that is, as you said, not fair by design.

---

> ### Author Response · Authors · 2023-11-20
> **Clarification by authors**
>
> Thank you for your additional feedback. We tried to keep our previous responses as concise as possible, so we apologize if they were not sufficiently clear. In the following, we provide further clarifications.
>
> 2. Regarding the advantage function, by its definition $A^\pi(s, a) = Q^\pi(s, a) − V^\pi(s)$ we get that optimal advantage value functions are indeed non-positive. However, in practice (and during training), we typically use function approximators with linear outputs to learn either $A^\pi(s, a), Q^\pi(s, a)$, or $V^\pi(s)$. As such, these functions can assume positive values during training but have non-positive values when converging to the optimal functions.
>
> We remark on two widely-applied examples in the literature that exploit positive advantage value functions, to further motivate why this could help in practice:
> - In value-based algorithms, dueling networks showed how learning $V^\pi(s), A^\pi(s, a)$ separately, significantly improves sample efficiency and performance. In particular, authors show that forcing the advantage function estimator to have zero advantage at the chosen action (i.e., the locally-optimal action at that step of training), has significantly lower performance than having positive advantages as the latter increases the stability of the optimization [A].
> - In policy-gradient algorithms, a step in the policy gradient direction should increase the probability of better-than-average actions and decrease the probability of worse-than-average actions. To this end, when using the advantage value function to measure whether or not the action at step $t$ is better or worse than the policy’s default behavior. We have that the gradient term $A^\pi(s_t, a_t)\nabla_\theta \pi_\theta(a_t|s_t)$ points in the direction of increased $\pi_\theta(a_t|s_t)$ if and only if $A^\pi(s_t, a_t) > 0$ [B, C].
>
> For this reason, we formalized our advantage-based version of Mac-IGM to also consider positive advantages. While this is not relevant for applying previous factorization methods to macro-actions, it would allow future research to build on existing methods (e.g., dueling networks) to design more efficient and scalable algorithms.
>
> 3. Regarding the results, they are key to support the claims of our paper.
>     - The fact that the performance "ranking" of the methods follows the primitive-action case, shows that our formalization based on MacDec-POMDPs is sound and works well in practice.
>     - They demonstrate the importance of the temporal-consistency problem that has not been considered by macro-action literature before. In particular, introducing the joint macro-state is a simple yet powerful way to address such an issue.
>     - These results also serve to compare previous macro-action-based works, showing that (i) it is possible to design principled value factorization methods for asynchronous Multi-Agent Reinforcement Learning (MARL), (ii) these CTDE value-based algorithms perform better than the current asynchronous MARL.
>     - Considering the superior performance over existing methods, our results set a new baseline for future CTDE macro-action-based works.
>
> 5) There are still some misunderstandings about the primitive-action environments. In particular, **an asynchronous task with unitary action duration is a primitive task** (we never refer to such setups as an asynchronous task with unitary action, since unitary actions make agents synchronous, or primitive). As such, there is no difference between primitive environments/algorithms and their unitary asynchronous counterparts (since the latter are simply primitive environments/algorithms). The reason why we reported results for the primitive-action factorization methods is twofold:
> - These results remark on the complexity of the standard benchmark scenarios employed by the macro-action literature. In particular, the original (primitive) approaches (VDN, QMIX, QPLEX) fail at solving even the easiest setups (as discussed in our response to Reviewer mc14).
> - We further motivate the importance of employing macro-action-based methods  as they successfully scale to higher-dimensional problems with a higher number of agents with different roles.
>
> We hope these additional comments helped to clarify some of your doubts and again, we appreciate your engagement and comments that we believe allowed us to improve our work. We remain available to further discuss our manuscript during these last days of the discussion period.
>
> [A] Z. Wang et al., “Dueling Network Architectures for Deep Reinforcement Learning,” ICML, 2016.
>
> [B] J. Schulman et al., "High-dimensional continuous control using generalized advantage estimation," ICLR, 2016.
>
> [C] J. Schulman et al., "Proximal Policy Optimization Algorithms," arXiv, 2017.

---

> > ### Comment · Reviewer_eKQs · 2023-11-21
> > **Reply to authors**
> >
> > I appreciate the additional clarification from the authors,as these proved helpful to clarify some of my concerns.
> >
> > 2. I see what your point here is, but from the paper the idea of letting the advantage get positive value seem to be presented as a contribution, rather than simply as an "hack" to improve learning. You should clarify that this has nothing to do with the learned decomposition, but it is rather an optimization trick that is used in accordance to the existing literature.
> > 3. I appreciate the results shown by your algorithms are to show the benefits of using asynchronous machinery and appropriate state/action representations, and that these can serve as a basis for for future directions.
> > 5. Also, I see why you are reporting results for primitive algorithms. What I was saying is that I would have expected the results for the asynchronous algorithms with unary time duration would have matched these, while indeed they were not. I think it would have been nice to show that indeed (as you claim in the paper), your proposed techniques reduces to the primitive versions with unary duration (and thus show that their performances match).
> >
> > I do not have any further question or remark for the authors, I will keep my score as it is (after having taken into account also replies to other reviewers), as it is already a positive one..

---

> ### Author Response · Authors · 2023-11-21
> **Thank you!**
>
> We would like to thank the reviewer for the constructive discussion that helped us improve our original submission. We will incorporate your last suggestions in the next paper revision, and remain available to clarify any other questions that might arise before the end of the discussion period.
>
> Best Regards,
>
> The Authors

---

### Official Review · Reviewer_mc14 · 2023-10-25

**Soundness:** 3 good
**Presentation:** 3 good
**Contribution:** 2 fair
**Rating:** 5
**Confidence:** 4

**Summary:**

In this paper, the authors delve into the domain of value factorization in asynchronous MARL. They extend the concept of IGM to accommodate macro-action settings and integrate it with pre-existing value factorization techniques. Furthermore, the paper offers insights and strategies on leveraging centralized information. Empirical experiments conducted on simple scenarios underscore the efficacy of the proposed method.

**Strengths:**

The problem of value decomposition for asynchronous MARL is interesting and open.
The paper is well organized and easy to read.

**Weaknesses:**

The novelty and contributions of this paper appear somewhat limited.
The paper introduces the concept of IGM for macro-actions. However, it's worth noting that this introduction may seem less impactful, as existing value factorization algorithms can inherently fulfill this requirement as long as state and action representations are adjusted in accordance with prior research, which is precisely what the paper demonstrates.
The experimental scenarios employed in this study are simplistic.

**Questions:**

1. The paper emphasizes that defining Mac-IGM and MacAdv-IGM is not trivial. But it seems that the primary distinction lies in considering only terminated macro-actions. Are there other complexities that merit attention?

2. It is apparent that extending existing value factorization methods to macro-action settings would inherently adhere to Mac-IGM. What is the significance of introducing Mac-IGM, and does it offer advantages in the development of algorithms, especially in cases where a direct extension might violate Mac-IGM? Further clarification on this aspect would enhance the paper's contribution.

---

> ### Author Response · Authors · 2023-11-13
> **Rebuttal by Authors**
>
> We thank Reviewer mc14 for acknowledging the interesting nature of asynchronous MARL. However, we believe several misunderstandings hindered the evaluation of our work; we clarify these below.
>
> - Regarding the impact of IGM on macro-actions, we believe the claim that *“existing value factorization algorithms can inherently fulfill this requirement as long as state and action representations are adjusted in accordance with prior research”* is severely misleading. In the following, we clarify our point of view, which also addresses the two reviewer's questions.
> **Without Mac-IGM and MacAdv-IGM, existing value factorization algorithms with adjusted state and action representations would not satisfy the action selection consistency**, since *Mac-IGM and MacAdv-IGM must consider what happens at the centralized level and decentralized level, separately*. For example, while the latter follows from the definition of a macro-action (i.e., agent $i$) selects a new macro-action based on the termination condition $\beta_i$ of the low-level policy of the previously executed macro-action), the former requires the conditional operator discussed in Eq. 5. *Moreover, MacAdv-IGM relaxes the assumptions of Adv-IGM* on the non-positive advantage values, while still guaranteeing consistency in the action selection as shown in the proof provided in Appendix A. Finally, *the formalization of Mac-IGM and MacAdv-IGM also allowed us to draw conclusions about the relationships over the primitive case*. In particular, we demonstrated how they represent more general classes of functions with respect to their primitive counterparts. It is worth *remarking on the temporal consistency problem* we discussed in Section 3.2 as naive implementations of asynchronous value factorization methods considering the actual state of the environment introduce significant noise in the training process, leading to policies that fail at solving even the most simple tasks (Fig. 5).
>
> - Regarding the experimental scenarios, the reviewer’s claim *“The experimental scenarios employed in this study are simplistic.”* also deems scarce attention to our experiments and limited knowledge of the asynchronous MARL literature. *We used most of the environments proposed in the latter, including a more complex version of the capture-target domain*. In addition, our response (as well as previous macro-action works such as Xiao et al., 2020a) shows that previous **synchronous, primitive approaches fail to solve even the most simple setup (BP-10). This is clear evidence of the complexity of these tasks that, we remark, represent the standard benchmark in the asynchronous MARL literature**. The complexity arises from a high degree of partial observability, combined with strong coordination requirements; characteristics that represent a serious challenge for existing MARL methods.
>
> As demonstrated by our responses to the other reviewers, we take all the feedback into serious consideration as they can help improve our manuscript. As such, we look forward to engaging in a fruitful discussion with Reviewer mc14 to further clarify any doubts regarding the proposed theory or methodology contributions.

---

> > ### Comment · Reviewer_mc14 · 2023-11-21
> >
> > Thanks for the detailed response.
> >
> > My primary concern still remains on understanding the specific advantages that the introduction of Mac-IGM offers in algorithm development. I am particularly interested in situations where a direct extension might violate Mac-IGM. By "direct extension", of course, I am not only referring to modifications to the state and action but also to other aspects considered by previous works, such as the update rule in Eq. 5. In seeking clarification, I would appreciate the author's emphasis on the methods themselves rather than attempting to address potential loopholes in my phrasing. Specifically, I am eager to understand whether Mac-IGM provides any guidance in enhancing the "naive implementations of asynchronous value factorization methods."
> >
> > Furthermore, the absence of proposed new benchmark environments and the perceived lack of novelty in the incremental aspects contribute to my uncertainty about the significance of this paper's contribution.

---

> > > ### Author Response · Authors · 2023-11-21
> > > **Clarification by authors**
> > >
> > > We appreciate your additional feedback. We hope that our efforts to clarify our point of view exhaustively, will not be misinterpreted as attempting to address potential loopholes in the raised doubts.
> > >
> > > The short answer is that Mac-IGM (similar to the primitive IGM) does not provide any guidance in enhancing implementations as these principles only formalize an action selection consistency requirement. For example, IGM does not give insights on using a monotonic mixer as in QMIX to have consistent action selection (the same holds for Mac-IGM).
> > > Conversely, Adv-IGM proposed in the QPLEX paper (Wang et al., 2016) suggests a fixed decomposition scheme for the design of Adv-IGM-based algorithms. In contrast, MacAdv-IGM relaxes such decomposition by allowing arbitrary learning of advantage and state (history) value functions. We discuss the benefits of MacAdv-IGM in our response to Reviewer eKQs. In addition, we drew the relationship between the class of functions representable by the primitive and macro-action-based formalization, which is also part of our first contribution regarding \{Mac/Mac-Adv\}-IGM.
> > >
> > > The state and action modifications and their effects on the centralized buffer are the main changes that allow AVF-based methods to work (as discussed in the pseudocode in Appendix B). Despite the claimed lack of methodological novelty, value factorization in the asynchronous framework would not work without these (as confirmed by the extended ablation study). Our experiments, in fact, show that "direct extensions" (i.e., using the environments' state) satisfy Mac-IGM, but fail at learning good policies due to the noise introduced by a temporal inconsistent state (as shown using the Implicit Function Theorem in Section 3.2). Overall, the macro-action literature has not considered factorization so far, nor the asynchronous nature of extra information. We believe finding simple yet effective solutions for addressing these issues is a benefit of our work and not a limitation.
> > >
> > > In this direction, we want also to point out two important things:
> > >
> > > - Methodological modifications on state/actions space have led to widely considered state-of-the-art multi-agent reinforcement learning (MARL) algorithms (for instance, MAPPO [D] is simply a decentralized PPO algorithm with a centralized critic). In this vein, while our practical modifications could be perceived as relatively simple, the empirical impact of our work is supported by a wide variety of experiments run over a high number of independent runs. This confirms their statistical significance [E], the correctness and impact of the proposed AVF-methods.
> > >
> > > - We also emphasize that our work aims to lay the groundwork for future research on asynchronous value factorization, and is not proposing a novel factorization methodology. In fact, regarding Eq. 5, we explicitly claim in Section 3.2 that "AVF-based algorithms are trained end-to-end to minimize Eq. 14, which resembles the fully centralized case of Eq. 5". The paper's contribution, in this case, lies in the fact that Eq. 14 uses history-state-based macro-action value functions due to the use of a mixing network, which gives significant performance advantages over Eq. 5. This aspect has been clarified in the previous paper revision, where we included results for the fully centralized method that use Eq. 5.
> > >
> > > Finally, regarding the lack of new benchmarks, we did show a 10-agent version of CT as existing macro-action-based methods already solved the original CT-2. Nonetheless, setting new benchmark environments was not the scope of our paper. We believe the field of asynchronous MARL should first focus on developing principled methods to solve existing well-known tasks as there is still work to do before achieving optimal performance in existing domains.
> > >
> > > Overall, we think all the updates to our manuscript have significantly clarified and improved our contribution, as also recognized by Reviewer JQAT. Even though our visions on (asynchronous) MARL literature can be different, we really appreciate your feedback as we are sure it will help us to improve our future works. We remain available to answer any other doubt that might arise before the discussion period ends.
> > >
> > > [D] C. Yu et al., "The Surprising Effectiveness of PPO in Cooperative, Multi-Agent Games," in NeurIPS 2022.
> > >
> > > [E] P. Henderson et al., "Deep Reinforcement Learning that Matters," in AAAI, 2018.

---

### Official Review · Reviewer_JQAT · 2023-10-31

**Soundness:** 3 good
**Presentation:** 3 good
**Contribution:** 2 fair
**Rating:** 6
**Confidence:** 4

**Summary:**

This paper introduces a value decomposition method for asynchronous multi-agent reinforcement learning (MARL). The authors define conditions called Mac-IGM and MacAdv-IGM, extending the well-known IGM condition to the asynchronous MARL setting. The proposed method, named AVF, combines local value functions with temporally consistent macro-states to alleviate noise from others' macro-action terminations. AVF can be integrated with various value decomposition methods, such as VDN, QMIX, and QPLEX. AVF combined with these algorithms, outperforms other asynchronous MARL algorithms that do not utilize value decomposition.

**Strengths:**

1. This paper is well-written with good readability and comprehensibility.
2. This paper introduces the conditions for value decomposition in an asynchronous MARL setting for the first time.
3. This paper presents a simple yet effective method to enhance performance and stability, which can be easily applied to other value decomposition methods with minimal modifications.

**Weaknesses:**

1. It appears that Mac-IGM and MacAdv-IGM are natural extensions of IGM, and the following propositions hold in a straightforward manner.
2. Additionally, while temporal consistency is essential for successful and stable learning, the method of utilizing temporally consistent states may lack novelty.

**Questions:**

1. [About Weakness 2] Have you experimented with other methods that utilize temporally consistent states? If so, could you provide the results and explain why these approaches have an advantage over others?
2. Could you please provide more results of Mac-IAICC on other environments that have already been considered? It would be greatly appreciated if you could include learning curves.

---

> ### Author Response · Authors · 2023-11-13
> **Rebuttal by Authors**
>
> We thank Reviewer JQAT for the comments about our work that helped us improve our manuscript, as shown in the updated submission. In the following, we summarize the papers’ modifications that address the reviewer’s doubts.
>
> - Regarding Mac-IGM and MacAdv-IGM, they *are novel formalizations that ensure the consistency between local and global macro-action selection*. This **is not straightforward since it takes into account the temporally extended nature of macro-actions that require considering the local and global cases separately**. During centralized action selection, the joint value has to condition on ongoing macro-actions $m^-$ to avoid biases in the joint estimation, while decentralized action selection requires consistent macro-action selection for only $|M^i|$ agents (i.e., agents whose macro-action is over). Moreover, Mac-AdvIGM differs from the original (primitive) formalization since it does not enforce the decomposition from Q-values as in (J. Wang et al., 2021). We believe this aspect has been significantly overlooked by the MARL literature, and has the potential to allow learning the value of a state (or history) separately. Such a decomposition has been shown to significantly improve performance in single-agent setups based on dueling networks [A]. However, this goes beyond the scope of our original submission, and we intend to explore this in future work. In addition, **we further extend the Mac-IGM and MacAdv-IGM contribution by drawing the relationships between the classes of functions that can be represented by the primitive and asynchronous IGM principles**. Such a result is formalized in Proposition 3.5, where we *show that the macro-action principles are more general versions of the primitive ones as they can represent value functions on both primitive and temporally extended action spaces*. It follows that AVF implementations of existing primitive baselines represent a broader class of functions over the primitive counterpart.
>
> - Regarding temporal consistency, the proposed method is a simple yet effective mechanism that has been successful in achieving good performance and stable learning. We discussed three different representations to deal with the temporal consistency problem and provided an ablation study showing the crucial role of the proposed mechanism (which has been significantly extended in the paper revision. Please, refer to the new Fig. 5 for further details.). Given the novelty represented by asynchronous MARL in the literature, **we are not aware of any other method that raises the problem of utilizing temporally consistent states in the mixer, nor other solutions for tackling such a novel problem**. Hence, *we kindly ask the reviewer to provide any reference that will allow us to perform additional experiments with other methods that utilize temporally consistent states*.
>
> - Regarding **Mac-IAICC, we performed an additional set of experiments (20 independent runs) on CT-10**, confirming the challenges posed by these environments for the policy-gradient baseline. To further strengthen our claims on the superior performance of AVF methods, **we also added a comparison with the fully centralized Cen-MADDRQN** (Xiao et al., 2020a) in the most complex setups (i.e., BP-30, WTD-T1, CT-10). Please, refer to the revised manuscript (Appendix E and the revised Table 1) and our answer to Reviewer E8n9 for further details about these results. Finally, as requested by the reviewer, we included the learning curves for both Mac-IAICC and Cen-MADDRQN in the supplementary material.
>
> Thanks to the reviewer’s insightful comments, we believe *the additional experiments and our clarifications have provided significant added value to our original submission* that we hope to see reflected in our papers’ evaluation. Nonetheless, we look forward to further clarifying any remaining doubts about our work during this discussion period.
>
> [A] Z. Wang et al., “Dueling Network Architectures for Deep Reinforcement Learning,” ICML, 2016.

---

> > ### Comment · Reviewer_JQAT · 2023-11-21
> > **New comment**
> >
> > I'm grateful for the comprehensive feedback from the authors. The newly introduced Figure 5 seems to provide somewhat confident evidence regarding the effectiveness of temporally consistent states. Additionally, the experimental results clearly show that AVF methods outperform other approaches significantly. Nonetheless, I still have concerns about the novelty of Mac-IGM and MacAdv-IGM. Since all of my concerns, except for those related to Mac-IGM, have been addressed during the discussion phase, I'm inclined to raise my score to 6.

---

> > > ### Author Response · Authors · 2023-11-21
> > > **Thank you!**
> > >
> > > Thank you for your comments that allowed us to improve our manuscript, and for recognizing the merits of our revised manuscript. The discussion period is close to its end, but we remain available to clarify any further questions that might arise before then.
> > >
> > > Sincerely,
> > >
> > > The Authors

---

### Official Review · Reviewer_E8n9 · 2023-11-01

**Soundness:** 2 fair
**Presentation:** 3 good
**Contribution:** 3 good
**Rating:** 5
**Confidence:** 4

**Summary:**

Authors introduce value factorization methods which were designed for synchronous framework to the asynchronous framework. Firstly, they formalize the new IGM principles for the asynchronous framework, proving they generalize the primitive cases. Secondly, they propose AVF algorithms which extend previous factorization methods to use macro-actions. Finally, authors verifies the performance of AVF algorithms through experiments in standard benchmark environments in macro-action literature, and explores the influence of using different extra information in the mixer.

**Strengths:**

In general, the content of the paper is well structured and organized.

1. Authors extend current popular principles to macro-action based settings: IGM and Advantage-based IGM, and provide extensive theoretical analysis of the proposed principles.

2. Authors provide empirical results in different environments.

3. The paper is easy to follow.

**Weaknesses:**

1. Dec-MADDRQN is a fully-decentralized value-based method, I don't think its performance is good enough to act as a baseline to compare with CTDE-based algorithm, perhaps you can compare with Mac-IAICC in more environments and show the results.

2. The paper lacks the pseudo-code of the training process of AVF algorithms.

3. For fairness, authors compare AVF algorithms with primitive value factorization methods in the primitive action version of BP-10, but they didn’t provide any explanations about what changes have been made in the primitive action version of BP-10 from the original version.

4. Regarding the use of extra information in the mixer, the authors list three different mechanisms, but the second mechanism that using the macro-state of the agent whose macro-action has terminated is missing in the experiment.

**Questions:**

**Major questions**

1. As shown in Figure 2, if the macro-action of agent at the current timestep is not terminated, its macro-observation will not change. But I think the reality is that a change in macro-action of any other agent will lead to a change in the macro-state, and ultimately lead to a change in the macro-observation of agent i. Can you explain your ideas about this?

2. In the field of macro-action based MARL, are there other CTDE value-based algorithms?

3. The authors mention low-level policy in the Preliminaries, but they do not mention how to select actions for low-level policy in AVF algorithms. In my opinion, if low-level policies do not work during the duration of a macro-action, then the macro-action can be simulated by executing a specific micro-action continuously over a period of time. Is it right?

4. Different agents’ macro-actions start and end at different time steps, how to choose the duration time $\tau$ of the joint macro-action?

5. As far as I know, in the value factorization algorithms, in addition to the IGM and Adv-IGM principles, [1] proposes DIGM principle and [2] proposes RIGM principle. Can you discuss whether and how these two principles can be applied to the asynchronous framework?

**Minor questions**

1. Formula (5) is missing $,$ between $m^{-}$ and $ \hat{o}^{\prime}$.

2. In the tuple of the Dec-POMDPs, the last element in the tuple should be $\gamma$, not $z$.

3. What is the measure compared in Table 1? Average win rate or average return?

**References**

[1] Wei-Fang Sun, Cheng-Kuang Lee, and Chun-Yi Lee. DFAC Framework: Factorizing the Value Function via Quantile Mixture for Multi-Agent Distributional Q-Learning. ICML 2021.

[2] Siqi Shen, Chennan Ma, Chao Li, Weiquan Liu, Yongquan Fu, Songzhu Mei, Xinwang Liu, and Cheng Wang. RiskQ: Risk-sensitive Multi-Agent Reinforcement Learning Value Factorization. NeurIPS 2023.

---

> ### Author Response · Authors · 2023-11-13
> **Rebuttal by Authors**
>
> We greatly appreciate the insights provided by Reviewer E8n9, which helped us to significantly strengthen our work, as shown in our updated submission. In the following, we clarify the main modifications and answer the reviewer’s questions.
>
> Weaknesses:
>
> 1. Regarding the baselines, we compare our best-performing algorithm (AVF-QPLEX) with **additional experiments** (20 independent runs as in the other tasks) **considering Mac-IAICC and Cen-MADDRQN** (the fully-centralized value-based method of (Xiao et al., 2020a)) in all the hardest tasks: BP-30, WTD-T1, CT-10. In CT-10, Mac-IAICC and Cen-MADDRQN perform poorly due to the higher number of agents and the use of centralized estimators. For similar reasons, Cen-MADDRQN also has lower performance than AVF-QPLEX in WTD-T1 but obtains higher results in BP-30 (due to the small size of the joint observation). Due to space limitations, we kept the CT-10 results in the main paper as Table 1, but we moved the experiments in WTD-T1 and BP-30 in Appendix E.
>
> 2. We **added the general pseudo-code of AVF algorithms**, along with an additional explanation, in Appendix B.
>
> 3. Regarding the primitive baselines, **we used the primitive version of BP-10 presented in the original Dec-MADDRQN paper** (Xiao et al., 2020a). In summary, agents can only perform 1-step movement and push actions. We included an exhaustive discussion of the primitive version in Appendix D.1.
>
> 4. Concerning the extra information in the mixer, we did not include the second mechanism (i.e., using the macro-state of the terminated agent) as it showed significantly lower performance than using the joint macro-state in our preliminary experiments (due to the introduced noise for the other agent values). Nonetheless, we **significantly extended our ablation study** (Figure 5), discussing the lower performance of AVF-{QPLEX, QMIX} based **on the second mechanism and the real state**, both in WTD-S and (newly) in BP-10. Please, refer to the new Fig. 5 for further details.
>
> Questions:
>
> 1. Regarding the macro-observations, it is true that a change in the macro-action of any agent will influence the macro-observation of agent i. However, following the MacDec-POMDP formalization (Section 2, Amato et al., 2019), *agent $i$ can only observe such influence upon the termination of its own macro-action $m^i$ when it receives its updated macro-observation*. Conversely, if agents receive a new macro-observation and select a new macro action every time any agent terminates a macro-action, they will likely converge to a primitive action-based solution (this aspect has also been discussed by previous macro-action works).
>
> 2. To the best of our knowledge, *there are no principled CTDE value-based algorithms in the field of macro-action-based MARL*, which further motivates the contributions of our work.
>
> 3. *It is correct that a macro-action can be simulated by executing a 1-step for a longer period*. For example, the “push” macro-action in BP domains is simply a repeated 1-step move forward action.
>
> 4. The *duration time $\tau$ of the joint macro-action* is used for the joint reward accumulation and *is set to the time interval between any two macro-action terminations*. We clarified this in Section 2.2.
>
> 5. While it would be interesting to *apply DIGM and RIGM* to the asynchronous framework, we believe it is out of the scope of our work. In particular, *it would also require formalizing distributional macro-action-based MARL algorithms*, which have not been currently investigated by the few published works considering asynchronous MARL.
>
> Regarding minor questions, we appreciate the attention reserved by the reviewer for our manuscript. We fixed Eq. 5 and the Dec-POMDP tuple. Moreover, Table 1 reports the average return as all the other experiments (but it can be also interpreted as an average win rate given the nature of the reward function in CT-10). We clarified this in the table’s caption.
>
> We believe our responses and the additional experiments provide a significant added value to our original submission that we hope to see reflected in the reviewer’s evaluation. Nonetheless, we look forward to further clarifying any remaining doubts about our work during this discussion period.

---

> > ### Comment · Reviewer_E8n9 · 2023-11-22
> >
> > I appreciate the response and clarifications made by the authors. Most of my questions have been addressed. However, some of my concerns still remain.
> >
> > Firstly, based on the author's explanation of the primitive version of BP-10, I believe that a macro-action task can be converted to a micro-action version by constraining one-step action, thus current primitive value factorization methods can be applied to solve it. Therefore, I think the authors should include more comparisons in a wider range of experimental environments' primitive versions against more advanced primitive value factorization methods.
> >
> > Secondly, according to the author's response to my first question, I think the treatment of local observation in the primitive case is more reasonable, which contributes to my uncertainty about the contribution of this paper.
> >
> > Lastly, in Section 3.1 of the paper, where it's stated that "Mac-IGM represents a broader class of functions over the primitive IGM", the proof only demonstrates a forward inclusion relation. It would be more convincing if the authors provided proof of the non-inclusion in the reverse direction, or if they could list examples that satisfy primitive IGM but not satisfy Mac-IGM.

---

> > > ### Author Response · Authors · 2023-11-22
> > > **Clarification by authors**
> > >
> > > We appreciate your additional feedback that allowed us to further improve our work. In the following, we provide further clarifications, referring to the updated manuscript for additional details.
> > >
> > > 1. It is generally correct that a macro-action task can be converted to a primitive action one by constraining one-step action. We included only BP-10 in our manuscript as a representative example, given that we obtained similar considerations in the other setups. Nonetheless, we agree this was not clear in our previous submission. Following your comment, we further revised our manuscript (in particular, Appendix D and the description in the main paper, Section 4) to include additional results in WTD-S, CT-10. Overall, these results confirm the limited performance of primitive factorization methods in these challenging tasks characterized by a high degree of partial observability and requiring strict cooperation.
> > >
> > > Despite employing QPLEX, which is one of the most competitive factorization approaches, we will take into account the reviewer's comment to include more baselines in the final version of the manuscript. However, we are sure the reviewer recognizes that asking for new baselines on the last day of the discussion period does not allow us to extend our broad experimental section with additional methods for both the asynchronous and primitive cases. Nonetheless, following our discussion with Reviewer eKQs, it is important to note that our contribution is orthogonal to the primitive case, and AVF can be potentially applied to arbitrary baselines. Overall, we believe the current set of experiments gives sufficient credit to our claims and shows the promise of asynchronous value factorization.
> > >
> > > 2. Regarding the treatment of local observation, we are not sure what the reviewer is claiming and we kindly ask for further clarification. In particular, how macro-observations are handled comes from the MacDec-POMDP framework, which is a well-established framework for asynchronous MARL. How macro-observations are updated is standard and the same method has been widely employed by previous macro-action-based works [F, G, H, I]. As such, we do not understand how this could foster your uncertainty regarding our work since it does not represent a contribution, nor a limitation of our paper.
> > >
> > > 3. Regarding the request to "list examples that satisfy primitive IGM but not satisfy Mac-IGM", we assume the reviewer meant the opposite (i.e., show an example satisfy Mac-IGM and not IGM, since Mac-IGM represents a broader class of functions over the primitive IGM). We further clarified this in the paper's revision (Appendix A.1.1). The high-level idea comes from the fact that a macro-action space $\mathcal{M}$ contains at least one macro-action (by definition of a MacDec-POMDP [F]) plus the whole set of primitive actions $\mathcal{U}$. Hence, $|\mathcal{M}|>|\mathcal{U}|$. For similar reasons, the observation and macro-observation spaces typically have the same size. It naturally follows that $|\hat{\mathbf{H}}\times\mathcal{M}|>|\mathbf{H}\times\mathcal{U}|$ (i.e.,
> > > the domain over which primitive action-value functions are defined is smaller than the domain over which macro-action-value functions are defined). Hence, $F^{\text{IGM}} \subset F^{\text{Mac-IGM}}$.
> > >
> > > [F] C. Amato et al., “Planning with Macro-Actions in Decentralized POMDPs.” AAMAS, 2014.
> > >
> > > [G] M . Liu et al., “Learning for Multi-robot Cooperation in Partially Observable Stochastic Environments with Macro-actions.” IROS, 2017.
> > >
> > > [H] S. Omidshafiei et al., “Decentralized control of multi-robot partially observable Markov decision processes using belief space macro-actions.” The International Journal of Robotics Research, 2017.
> > >
> > > [I] Y. Xiao et al., “Asynchronous Actor-Critic for Multi-Agent Reinforcement Learning,” NeurIPS, 2022.

---

### Author Response · Authors · 2023-11-17
**Discussion period update**

Dear Reviewers,

Almost a week has passed since we submitted our paper revision and responses and, after the upcoming weekend, the discussion period will be close to its end. We would like to have enough time to appropriately address any further doubts the experiments you asked for might have raised. For this reason, we kindly ask you to engage in a meaningful discussion that, if needed, will help us further improve our work.

For simplicity, in the following we summarized the main changes we have made to answer your questions, referring to the individual responses for more details.

- We significantly extended our experiment section with:
    1. New experiments with Mac-IAICC (requested by Reviewer E8n9) along with the training curves for the existing experiments (requested by Reviewer JQAT).
    2. New experiments with a new centralized macro-action baseline (requested by Reviewer eKQs).
    3. New experiments in the ablation study with an additional domain and asynchronous factorization methods that use only one macro-state (requested by Reviewer E8n9).
All these new experiments come with an additional description of what is going on and why they confirm our previous claims.

- In addition, we carefully revised parts of the manuscript to address the reviewers' concerns:
    1. We introduced a general pseudocode and description for the proposed framework, clarified the primitive task setup, and fixed the typos (requested by Reviewer E8n9).
    2. We further discussed the merits of Mac-IGM and, in particular, MacAdv-IGM (requested by Reviewer mc14).
    3. Revising the background section and clarifying the practical advantage of MacAdv-IGM, and our comparison with primitive methods (requested by Reviewer eKQs).

Thanks to the reviewer’s comments, we believe these experiments and paper revisions have provided significant added value to our original submission which we hope to see reflected in the evaluation. We look forward to further clarifying any remaining doubts about our work during this discussion period.

Sincerely,

The Authors

---

### Meta-Review · Area_Chair_9Pd3 · 2023-12-04

**Metareview:**

This paper presents a novel method for multi-agent RL to introduce value factorization to the asynchronous framework.

**Reviewers have reported the following strengths:**
- Quality of writing;
- The method is sound and novel;
- Good empirical performance;

**Reviewers have reported the following weaknesses:**
- Some theoretical results are not properly justified by the empirical evidence;
- A wider range of experiments is needed.

**Decision**

The authors' rebuttal helped solve the Reviewers' doubts, However, concerns remained about the motivation and novelty of the method. This paper seems close to being in good shape for acceptance, but improvement in the justification of the proposed method is needed, especially from an experimental perspective. I encourage the authors to address the remaining Reviewers' concerns in a future submission.

**Justification For Why Not Higher Score:**

N/A

**Justification For Why Not Lower Score:**

N/A

---

### Decision · Program_Chairs · 2024-01-16

Reject